# *Arabidopsis* proteins with a transposon-related domain act in gene silencing

Yoko Ikeda[1,2,*], Thierry Pélissier[1,*], Pierre Bourguet[1], Claude Becker[3], Marie-Noëlle Pouch-Pélissier[1], Romain Pogorelcnik[1], Magdalena Weingartner[4], Detlef Weigel[3], Jean-Marc Deragon[5] & Olivier Mathieu[1]

Transposable elements (TEs) are prevalent in most eukaryotes, and host genomes have devised silencing strategies to rein in TE activity. One of these, transcriptional silencing, is generally associated with DNA methylation and short interfering RNAs. Here we show that the *Arabidopsis* genes *MAIL1* and *MAIN* define an alternative silencing pathway independent of DNA methylation and short interfering RNAs. Mutants for *MAIL1* or *MAIN* exhibit release of silencing and appear to show impaired condensation of pericentromeric heterochromatin. Phylogenetic analysis suggests not only that *MAIL1* and *MAIN* encode a retrotransposon-related plant mobile domain, but also that host plant mobile domains were captured by DNA transposons during plant evolution. Our results reveal a role for *Arabidopsis* proteins with a transposon-related domain in gene silencing.

[1] Université Clermont Auvergne, CNRS, Inserm, GReD, Clermont–Ferrand F-63000, France. [2] Institute of Plant Science and Resources, Okayama University, 2-20-1 Chuo, Kurashiki 710-0046, Japan. [3] Department of Molecular Biology, Max Planck Institute for Developmental Biology, Tübingen D-72076, Germany. [4] Molekulare Pflanzenphysiologie, Biozentrum Klein Flottbek, Universität Hamburg, Hamburg D-22609, Germany. [5] Laboratoire Génome et Développement des Plantes (LGDP), CNRS, UMR5096, Université de Perpignan Via Domitia, 58 Avenue Paul Alduy, Perpignan Cedex 66860, France. * These authors contributed equally to this work. Correspondence and requests for materials should be addressed to O.M. (email: olivier.mathieu@uca.fr).

In *Arabidopsis thaliana* genomes, silenced transposable elements (TEs) and repeats are enriched in pericentromeric heterochromatin, associated with dense DNA methylation in the three contexts CG, CHG and CHH (where H is any base but G), and have high levels of repressive chromatin marks such as histone H3 lysine 9 di- and lysine 27 monomethylation (H3K9me2, H3K27me1)[1–4]. METHYLTRANSFERASE1 (MET1) propagates CG methylation patterns[5], and while mechanisms maintaining CHG DNA methylation and H3K9me2 are intertwined[6], H3K27me1 patterns are maintained by *ARABIDOPSIS TRITORAX-RELATED PROTEIN 5* (*ATXR5*) and *ATXR6* independently of DNA methylation[7]. Non-CG DNA methylation and H3K9me2 also highly correlates with 24-nt siRNA accumulation[8]. The chromatin remodeler DECREASE IN DNA METHYLATION 1 (DDM1) plays a central role in regulating heterochromatin DNA methylation in the three cytosine contexts, likely by allowing DNA methyltransferases to access their targets[9,10].

Stable transcriptional gene silencing is achieved by the concerted action of multiple epigenetic mechanisms[11]. Among these, CG DNA methylation plays a pivotal role as reflected by the magnitude of transcriptional activation and the number of loci derepressed in mutants with low CG methylation levels such as *met1* or *ddm1* (refs 12,13). Non-CG DNA methylation, H3K9me2 and 24-nt siRNA production contribute additional layers of silencing at certain loci[8,12,14], and synergistic release of silencing occurs when deficiencies in maintaining these marks are introduced into genetic backgrounds with reduced CG DNA methylation[15–18].

Transcriptional gene silencing of selected, mainly heterochromatic loci, is further secured by silencing factors acting largely independently of DNA methylation. The best described are MORPHEUS' MOLECULE1 (MOM1), members of the *Arabidopsis* MICRORCHIDIA (AtMORC) ATPase family, notably AtMORC6, and the redundant H3K27 monomethyltransferases ATXR5 and ATXR6 (refs 7,19,20). *MOM1* and *AtMORC6* regulate silencing of some TEs, repetitive sequences and transgenes through distinct silencing pathways, with *MOM1* having the largest spectrum[21]. Although *AtMORC6* seems to contribute to heterochromatin condensation[19], the precise molecular mechanisms of action of *MOM1* and *AtMORC6* remain poorly understood. Transcriptional activation of many heterochromatic TEs and repeats in *atxr5 atxr6* double mutants is thought to be a consequence of reduced H3K27me1 levels, and it is associated with heterochromatin decondensation and heterochromatic overreplication[4,7].

Here, we describe a locus identified from a screen for gene-silencing mutants and show that the corresponding gene, *MAIL1*, and its close homologue *MAIN,* define an alternative silencing pathway that is largely independent of DNA methylation and siRNA production. *MAIL1* and *MAIN* encode a transposon-related plant mobile domain and appear to contribute to pericentromeric heterochromatin condensation.

## Results

**Mutations of *MAIL1* release gene silencing.** The *Arabidopsis* L5 line contains a model silenced locus of tandem-repeats of a β-glucuronidase (GUS) transgene, similar to heterochromatic repeats and TEs[22]. We carried out a mutant screen in the L5 background for reactivation of the L5 transgene. Besides new alleles of known DNA methylation factors, including *MET1* (ref. 5), *DDM1* (ref. 9) and HOMOLOGY-DEPENDENT GENE SILENCING1 (*HOG1*)[23] (Supplementary Fig. 1), we identified a new locus, *KUMONOSU* (Japanese for cobweb, abbreviated *KUN*) (Fig. 1a,b). In addition to the L5 locus, endogenous heterochromatic repeats and TEs were released from silencing in *kun* mutants (Fig. 1c). Mapping-by-sequencing indicated that

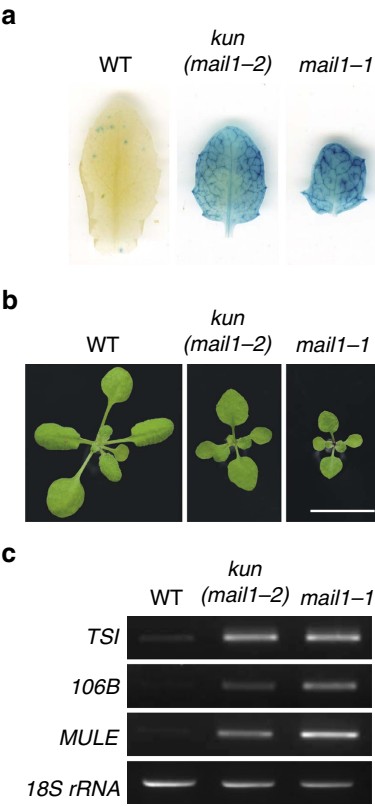

**Figure 1 | *mail1* mutations induce loss of silencing at the L5 locus and endogenous loci.** (**a**) WT plants carrying the L5 transgene showed no glucuronidase activity, while the L5 locus was strongly reactivated in leaves of *kun* and *mail1-1* mutants. (**b**) Photos of 3-week-old WT, *kun* (*mail1-2*) and *mail1-1* plants. Scale bar, 1 cm. (**c**) RT–PCR analysis of transcripts from the *TSI* and *106B* heterochromatic repeats and the *AT2TE28020* MULE. Amplification of 18S rRNA is shown as loading control.

*kun* contained a mutation in AT2G25010 (*MAINTENANCE OF MERISTEMS-LIKE1*, *MAIL1*; Supplementary Fig. 2a,b), reported to play a role in cell division and differentiation, especially in meristematic cells[24]. We confirmed a causal role of *MAIL1* in L5 silencing by crossing the transgene to an independent *mail1* knockout mutation (Fig. 1a), complementation test between *kun* and *mail1*, and rescue of *kun* defects with a *MAIL1* genomic copy (Supplementary Fig. 2c). Hence, *kun* was renamed *mail1-2*. Both the *mail1-1* knockout mutant and *mail1-2* plants grew more slowly than wild type (WT), with the growth of *mail1-1* plants being more delayed than that of *mail1-2* plants (Fig. 1b and Supplementary Fig. 3). There was no obvious cell death in *mail1-2* and in young leaves of either *mail1* mutant, but mature *mail1-1* leaves suffered from pronounced cell death around the vascular tissue[24] (Supplementary Fig. 3). We used the *mail1-1* null mutant[24] (hereafter referred to as *mail1*) for subsequent analyses.

To define the global effects of *mail1* on gene expression, we compared the transcriptomes of *mail1* and WT seedlings generated by RNA sequencing (RNA-seq). This analysis revealed that loss of *MAIL1* predominantly induced upregulation of transcripts, including at loci located within pericentromeric heterochromatin (Fig. 2; Supplementary Fig. 4a; Supplementary Data 1 and 2). Accordingly, upregulated loci contained many TEs that belonged to both the DNA transposon and retrotransposon classes (Supplementary Fig. 4b, Supplementary Data 1) and were enriched in pericentromeric heterochromatin

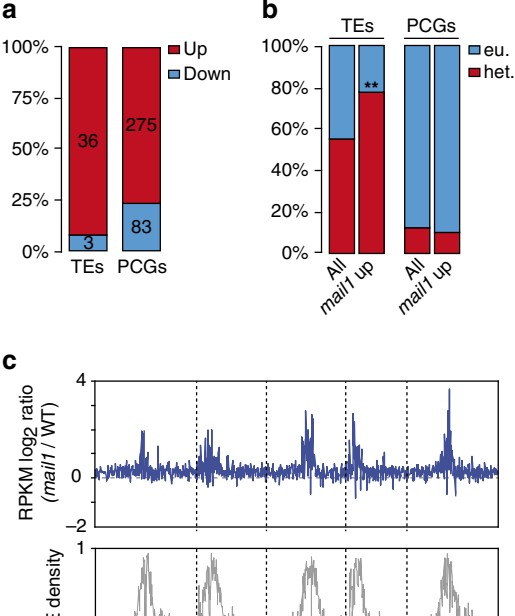

**Figure 2 | Mutation of *MAIL1* mostly induces upregulation of transcripts at both PCGs and heterochromatic TEs.** (**a**) Fraction and number of TEs and PCGs significantly up- (Up) or downregulated (Down) in *mail1* seedlings versus WT. (**b**) Fraction of TEs and PCGs significantly upregulated in *mail1* (*mail1* up) located in euchromatin (eu.) or in pericentromeric heterochromatin (het.). Pericentromeric heterochromatin coordinates were assigned based on the distribution of repetitive elements, PCGs and DNA methylation across chromosomes as in ref. 2. Asterisks mark statistically significant differences from the distribution of total *Arabidopsis thaliana* TEs and PCGs (All, $z$-score = − 2.68, $P < 0.01$). (**c**) Overview of the five *A. thaliana* chromosomes showing the log2 ratios (*mail1-1*/WT) of mean RPKM values in 100 kb windows. The lower panel shows TE density in 100 kb windows, demonstrating the close correspondence between TE density and increased transcript accumulation in *mail1-1* mutants.

(Fig. 2b,c). Such genomic distribution bias was not observed for differentially upregulated protein-coding genes (PCGs; Fig. 2b), which included the well-known epigenetically regulated genes *SUPPRESSOR OF DRM1 DRM2 CMT3* (*SDC*)[25] and the pericentromeric gene *QUA-QUINE STARCH* (*QQS*)[26]. Together, failure to maintain L5 transgene silencing, preferential upregulation of genes including *SDC* and *QQS* and de-repression of heterochromatic TEs in *mail1,* indicate that *MAIL1* is required for epigenetic silencing of a subset of genomic loci.

**MAIL1 acts independently of DNA methylation and siRNA**. We suspected that global upregulation in *mail1* mutants might be due to a reduction in DNA methylation, and therefore analysed genome-wide DNA methylation at single-nucleotide resolution in *mail1* and WT seedlings by bisulfite sequencing (BS-seq). Average methylation levels and global methylation patterns over all TEs and PCGs were similar in WT and *mail1* (Supplementary Fig. 5a,b). Likewise, DNA methylation was not changed at either PCGs or TEs upregulated in *mail1* (Fig. 3a). To further investigate a potential effect of *mail1* on DNA methylation, we determined differentially methylated regions (DMRs) between WT and *mail1*. As controls for DMR calling, we used previously published data to calculate DMRs in the *mom1-2* and *atmorc6-3* mutant backgrounds, known to have mostly

unchanged DNA methylation[19,21,27], and in the strongly hypomethylated *ddm1-2* mutant[27]. The total number and length of DMRs was similarly low in *mom1* and *atmorc6*, and it was in the same range or even lower in *mail1* seedlings (Supplementary Fig. 6a,b). In addition, hypomethylated DMRs and upregulated loci in *mail1* only poorly overlapped (Fig. 3b). Together, these data demonstrate that DNA methylation with functional consequences is largely unaltered in *mail1*.

The fact that silencing release in *mail1* was not accompanied by a loss of DNA methylation suggested that *MAIL1* controls gene silencing either downstream or independently of DNA methylation. To discriminate between these alternatives, we introduced *mail1* into the *ddm1-2* background and examined transcript levels from a subset of loci upregulated in *mail1*. The *MAIL1*-controlled loci were highly methylated in WT and showed decreased DNA methylation and release of silencing in *ddm1* (Fig. 4a and Supplementary Fig. 7). For all examined loci, including the *CACTA-like* and *MULE* TEs that are virtually unmethylated in *ddm1* (Supplementary Fig. 7), transcript accumulation was further enhanced in *mail1 ddm1* double mutants relative to either single mutant parent (Fig. 4a). Similarly, growing *mail1* plants on the DNA methylation inhibitor 5-aza-2′-deoxycytidine enhanced *mail1*-induced loss of silencing (Supplementary Fig. 8a). Taken together, the mostly unchanged DNA methylation in *mail1* and the synergistic effects of *ddm1* and *mail1* mutations on gene silencing, notably at *MAIL1*-target loci that totally lose DNA methylation in *ddm1*, strongly suggest that *MAIL1* functions independently, rather than downstream, of DNA methylation.

The biogenesis of 24-nt siRNAs, a central component of the RNA-directed DNA methylation silencing pathway, requires activity of the RNA-dependent RNA polymerase RDR2 (ref. 28). We created *mail1 rdr2* double mutants and found that expression of *mail1* target loci upregulated in *rdr2* was further enhanced in *mail1 rdr2* (Supplementary Fig. 8b). RNA gel blots using flower RNA at selected MAIL1-targetted loci and siRNA-producing loci suggested no changes in siRNA accumulation in *mail1* (Fig. 4b; Supplementary Fig. 9a). Quantitative PCR with reverse transcription (RT–PCR) analysis showed that MAIL1-targets were upregulated in both seedlings and flower tissues (Supplementary Fig. 9b). Small RNA sequencing of flower RNA confirmed that siRNA accumulation was not altered in *mail1* mutants (Fig. 4c,d and Supplementary Fig. 9c,d). Thus, we conclude that *MAIL1* enforces gene silencing through a pathway that is independent not only of DNA methylation, but also of siRNAs.

**MAIL1 function is mostly independent of MOM1 and AtMORC6.** *MOM1* and *AtMORC6* define two separate silencing pathways that are likely independent of DNA methylation[19,29]. Silencing of the L5 transgene is released in *mom1* mutants[20], and we found that L5 silencing is also alleviated in *atmorc6*, although to a lesser extent than in *mail1* (Supplementary Fig. 10a). Re-analysis of transcriptome data for *mom1* (ref. 12) and *atmorc6* (ref. 19) revealed limited overlap with *mail1*-overexpressed TEs (Supplementary Fig. 10b). Quantitative RT–PCR analyses revealed that combining *mail1* with either *mom1* or *atmorc6* mutations generally had a synergistic effect on silencing release (Supplementary Fig. 10c). These data indicate that *MAIL1*, *MOM1* and *AtMORC6* mediate gene silencing at sets of independent and common targets through largely independent molecular pathways.

**H3K27me1 and MAIL1-mediated silencing**. Immunocytology indicated that nuclear patterns of H3K9me2 and H3K27me1 were unaltered in *mail1* (Supplementary Fig. 11a). Chromatin

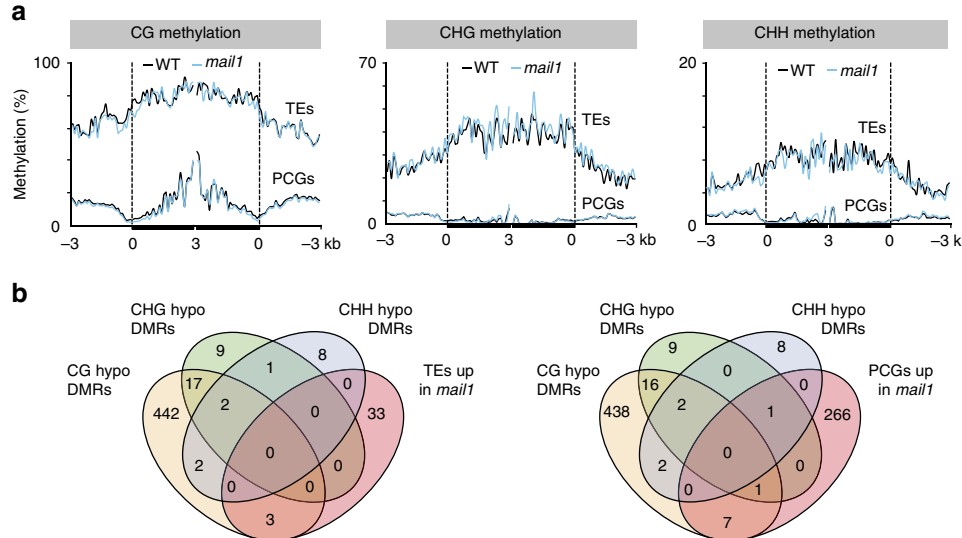

**Figure 3 | DNA methylation is not impaired in *mail1*.** (**a**) PCGs or TEs significantly upregulated in *mail1* were aligned at the 5′-end (left dashed line) or the 3′-end (right dashed line), and average methylation levels in CG, CHG and CHH sequence contexts for each 100-bp bin in WT and *mail1* were plotted from 3 kb upstream to 3 kb downstream. (**b**) Venn diagrams of overlap between genomic regions differentially hypomethylated in *mail1* (hypo DMRs) in the three cytosine sequence contexts and TEs and PCGs significantly upregulated in *mail1*.

immunoprecipitation (ChIP) revealed no significant changes in H3K9me2 at several *MAIL1*-regulated loci. This was very different from *ddm1-2* mutants, in which the same set of loci nearly lost all H3K9me2 marks (Supplementary Fig. 11b). This indicates that *MAIL1* does not control H3K9me2, which is in agreement with DNA methylation being unchanged in *mail1* mutants. In contrast, all loci examined had little H3K27me1 in *mail1*, in a similar range as observed in *atxr5/6* (Supplementary Fig. 11c). *ddm1-2* also showed drastic H3K27me1 depletion, which is at variance with previous inferences from non-quantitative analyses[3]. We further confirmed global H3K27me1 depletion in *ddm1-2* by immunocytology and protein blot experiments (Supplementary Fig. 12a,b).

Loss of silencing correlated with decreased H3K27me1 levels in *mail1*, and H3K27me1 was nearly lost at *MAIL1*-controlled loci in *ddm1-2* mutants. Because we had found that *MAIL1* and *DDM1* act synergistically in silencing, it is most likely that silencing defects in *mail1* are not a direct consequence of decreased H3K27me1 levels. To further confirm this hypothesis, we generated *mail1 atxr5/6* triple mutants and profiled transcriptomes of *atxr5/6* and *mail1 atxr5/6* seedlings by RNA-seq. Only a few TEs were activated in both *mail1* and *atxr5/6* mutants, but a large number of TEs became reactivated in the *mail1 atxr5/6* triple mutants (Supplementary Fig. 13a; Supplementary Data 1). ChIP analysis at several *MAIL1*-regulated loci revealed mostly similar H3K27me1 levels in *mail1, atxr5/6* and *mail1 atxr5/6* (Supplementary Fig. 13b). We conclude that the *MAIL1* and *ATXR5/6* silencing pathways seem to be largely non-overlapping, reinforcing the view that decreased H3K27me1 levels are likely not the direct cause of silencing release in *mail1*.

**Heterochromatin condensation appears impaired in *mail1*.** TEs and repeats upregulated in *mail1* preferentially localized to pericentromeric heterochromatin (Fig. 2b,c), which appear as chromocenters when nuclei are stained with 4′, 6-diamidino-2-phenylindole (DAPI). We observed increased chromocenter area relative to that of the entire nucleus in DAPI-stained *mail1* nuclei, suggesting potential chromocenter decondensation

(Supplementary Fig. 14a,b). Physical expansion of chromocenters in *mail1* was neither associated with increased nuclear size (Supplementary Fig. 14c) nor with endoreduplication defects (Supplementary Fig. 14d,e). Notably, unlike *atxr5/6* nuclei[4], *mail1* nuclei did not show over-replication of heterochromatin (Supplementary Fig. 14e). The *atxr5/6* heterochromatin over-replication phenotype was also not enhanced in *mail1 atxr5/6* (Supplementary Fig. 14e). To further investigate heterochromatin condensation in *mail1*, we performed fluorescent *in situ* hybridization on interphase nuclei using a probe for the 106B LTR-like pericentromeric repeats, which are transcriptionally upregulated in *mail1* (Fig. 1c). While most WT nuclei showed a fluorescent signal clustered around chromocenters, a higher proportion of *mail1* nuclei displayed signals dispersed within the nucleoplasm (Fig. 5a,b; Supplementary Fig. 15). Assuming equal DNA content, these data support a role of *MAIL1* in condensation of pericentromeric heterochromatin. MAIL1 is a nuclear protein[24], and analysis of a MAIL1-GFP fusion protein suggested that MAIL1 is not preferentially associated with chromocenters (Supplementary Fig. 16).

**MAIN and MAIL1 operate in the same silencing pathway.** In *Arabidopsis thaliana*, *MAIL1* is part of a small gene family comprising three other genes, *MAIL2, MAIL3* and MAIN-TENANCE OF MERISTEMS *(MAIN)*, which encode proteins with a so-called aminotransferase-like, plant mobile domain (PMD)[24,30]. MAIN, MAIL1 and MAIL2 are highly similar, while MAIL3 contains an additional serine/threonine phosphatase domain[24]. Analysis of mutants and RNAi lines did not suggest a role of *MAIL2* or *MAIL3* in gene silencing (Supplementary Fig. 17a). In contrast, loss of *MAIN* function released L5 silencing and led to upregulation of pericentromeric heterochromatin transcripts and apparent heterochromatin decondensation, similar to what we had seen in *mail1* mutants (Fig. 5; Supplementary Figs 14a,b and 17). There were no obvious additional defects in *mail1 main* double mutants (Fig. 5; Supplementary Figs 11, 14a,b,d and 17a,d,e), suggesting that both proteins act in the same pathway, perhaps as heterodimers, to compact chromatin, which in turn silences transcription.

**Phylogenetic analysis of PMDs.** Gene silencing is ensured by a diversity of pathways in plants. Since our results revealed a role of *Arabidopsis* PMD proteins in silencing, we sought to determine

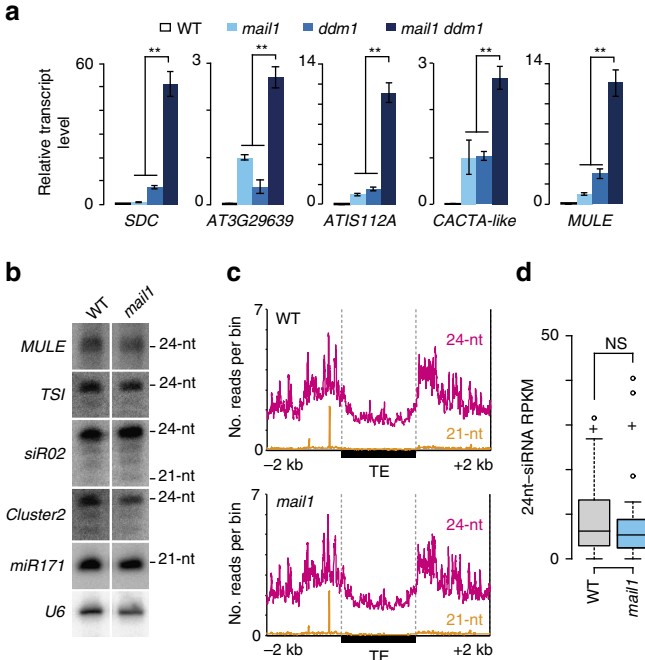

**Figure 4 | Loss of silencing in *mail1* is independent of DNA methylation and siRNA accumulation.** (a) Expression analysis of *mail1*-overexpressed loci by quantitative RT–PCR in the indicated genotypes. *mail1* levels are set to 1; values represent means from two to three biological replicates ± s.e.m. Asterisks mark statistically significant differences from the *mail1 ddm1* double mutant (unpaired, two-sided Student's *t*-test, $P < 0.05$). (b) RNA gel blot analysis of siRNA accumulation. U6 RNA is shown as loading control. (c) 21- and 24-nt siRNA levels at TEs are unaltered in *mail1*. Number of uniquely mapped 21- and 24-nt siRNAs plotted along all *Arabidopsis thaliana* TEs from 2 kb upstream to 2 kb downstream of the TE annotation. Upstream and downstream sequences were divided each into 2,000 one-nucleotide bins, and TEs were divided into 2,000 bins of equal length. Average siRNA levels were computed for each bin in WT and *mail1*. (d) Boxplot of 24-nt siRNA accumulation level at *mail1*-upregulated TEs. Each box encloses the middle 50% of the distribution with the center lines marking the medians. Crosses mark sample means. Whiskers extend 1.5 times the interquartile range from the 25th and 75th percentiles. Outliers are represented by circles. NS, nonsignificant (unpaired, two-sided Student's *t*-test, $P > 0.9$).

how widespread PMD proteins are in plants. The PMD was first described as a domain associated with a *MULE* element in rice (*Oryza sativa*)[30] and was later found associated with the *Ty3/Gypsy* group of LTR retrotransposons in several plant genomes[31]. We found that PMD proteins are ubiquitous in angiosperms, with some being encoded by regular PCGs, while others are encoded by TEs (Fig. 6 and Supplementary Fig. 18). A phylogeny of the PMD family indicated that the PMD-A1 and PMD-A2 clades are exclusively encoded by *Gypsy* retrotransposons that are largely distributed amongst angiosperm species. The two other clades, PMD-C, which includes the MAIN and MAIL proteins, and PMD-B, are mainly associated with PCGs in all angiosperms. There are, however, PMD-C members that occur as fusion proteins with transposases of *MULE* TEs in several grasses and in *Amborella trichopoda*, a species at the base of the angiosperms (Fig. 6). That the *MULE*-encoded PMDs are embedded within the PMD-C clade strongly suggests that they are derived from host genes and that these have been captured by the *MULE* TEs. Together, these results are consistent with a scenario where a *Gypsy*-encoded PMD (PMD-A) was present in the common ancestor of angiosperms in association with the PMD-B- and C-containing gene families, and more recently, genic PMD-C was captured twice independently by *MULE* DNA transposons.

## Discussion

Overall, this study demonstrates that *MAIL1* and *MAIN* define a silencing pathway that is independent of major repressive epigenetic marks, including DNA methylation, and suggests a role for these two genes in controlling proper condensation of pericentromeric heterochromatin. The molecular mechanisms by which *MAIL1* and *MAIN* may control chromatin condensation remain unknown. Analyses of genetic interactions between *mail1* and *main* mutations presented here suggest that MAIL1 and MAIN proteins might act in the same protein complex. The identification of MAIL1 and/or MAIN-interacting proteins, possibly part of such complex, will certainly provide more insights into how *MAIL1* and *MAIN* modulate chromatin compaction. Other silencing pathways, involving *MOM1* and *AtMORCs* are also likely to function largely in parallel to, rather than downstream of DNA methylation, and at least in the case of *AtMORC1* and *AtMORC6*, also influence chromatin condensation. Like for MOM1, which preferentially targets loci with intermediate heterochromatin properties for silencing[32], it is possible that AtMORCs and MAIL1/MAIN are required to maintain silencing of a subset of genomic targets exhibiting distinctive chromatin signatures. It would be interesting to

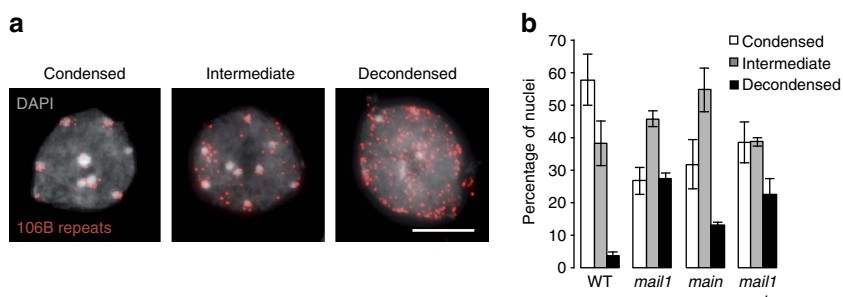

**Figure 5 | Condensation of pericentromeric heterochromatin appears impaired in *mail1* and *main* mutants.** (a) Representative images of the three chromatin condensation states observed in WT leaf interphase nuclei hybridized with a probe for 106B pericentromeric repeats and counterstained with DAPI. Representative images of mutant nuclei hybridized with the same probe are shown in Supplementary Fig. 15. Scale bar, 5 μm. (b) Proportions of nuclei belonging to each class of nuclei shown in a based on the degree of condensation of the 106B fluorescent signal in the indicated genotypes. About 100–220 nuclei were scored per genotype. Values represent means from two biological replicates ± maximum and minimum.

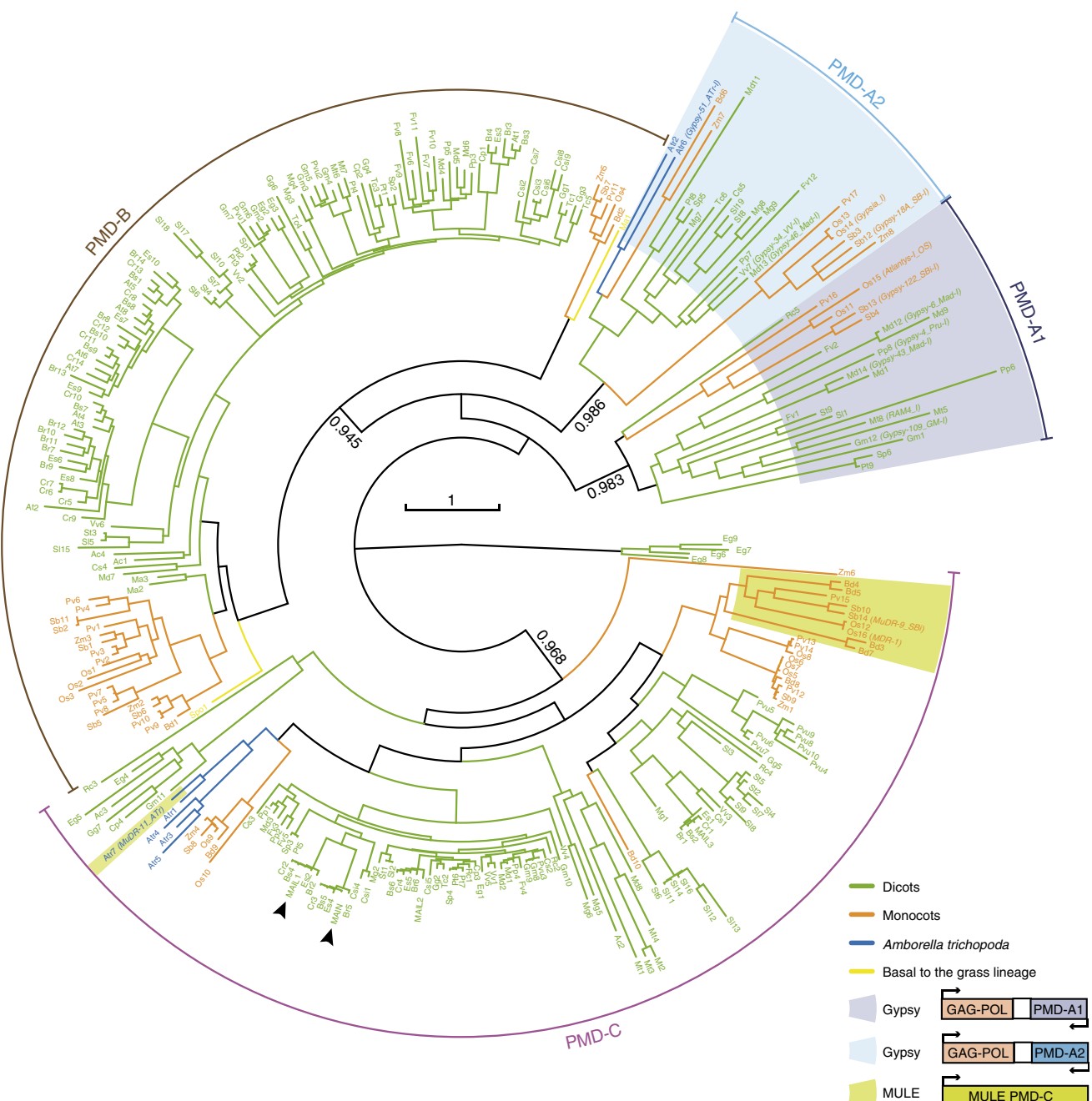

**Figure 6 | MAIL1 and MAIN contain a TE-related plant mobile domain.** A phylogenetic tree was built from 308 PMDs from 33 species representing the diversity of this motif in angiosperms (Supplementary Data 3). The three clades A, B, C are indicated. The species codes are given in Supplementary Table 1. The colour code is the following: green for dicotyledonous species, orange for monocotyledon species, blue for the basal *Amborella trichopoda* and yellow for the two species that are basal to the grass lineage. Names of the TE sequences extracted from the Repbase database are presented in parenthesis. PMDs associated with TEs are highlighted in violet, blue or green depending on their clade and a schematic representation of their organization is shown. Other PMDs are associated with non-TE PCGs. MAIL1 and MAIN proteins are indicated with arrow heads. Statistical supports of key nodes calculated with the approximate likelihood-ratio test are indicated. Scale bar indicates one substitution/site.

address this possibility by determining patterns of post-translational histone modifications at MAIL1/MAIN and AtMORCs targets.

Plant genomes have evolved remarkably complex mechanisms of epigenetic control. Both *MAIL1* and *MAIN* encode a conserved protein motif that shares a common ancestry with a retrotransposon-associated amino acid sequence. We propose two scenarios to explain this phylogenetic relationship: (i) a preexisting genic PMD was co-opted by a *Gypsy* retrotransposon; or (ii) a genic PMD was

domesticated from a *Gypsy*-associated PMD in the common ancestor of angiosperms, and subsequently evolved to acquire its silencing function. During evolution of eukaryotes, TE genes have been repeatedly recruited by their hosts and there are many instances of host genomes domesticating TE-encoded factors to serve important cellular functions[33–36]. The human centromeric protein CENP-B is derived from transposases of *pogo* DNA transposons[37] and in the fission yeast *Schizosaccharomyces pombe*, CENP-B homologues were shown to be involved in genome organization and in the surveillance

for retrotransposons[36]. Although given these precedents, we tend to favour the PMD 'domestication' scenario, leading to the evolution of an additional silencing pathway following a retrotransposon invasion, the question of the ancestry of PMD-A relative to the genic PMD-B and C remains fully open. Whatever the answer to this question might be, our analysis suggests that, in two recent and independent events, PMD genes have been captured by DNA TEs. *MULE* elements are able to capture genomic sequences, including gene fragments, in a process termed transduplication[38]. Although the function of the PMD domain itself remains to be determined, it is tempting to speculate that the captured protein is being used in a TE anti-silencing pathway by *MULE* elements.

## Methods

**Plant material.** Mutants *ddm1-2* (ref. 9), *rdr2-2* (SALK_059661) (ref. 28), *mom1-2* (SAIL_610_G01) (ref. 32), *atmorc1-4* (SAIL_1239_C08) (ref. 19), *atmorc6-3* (GK_599B06) (ref. 19), *atxr5 atxr6* (SALK_130607, SAIL_240_H01) (ref. 7), *main-2* (GK-728H05) (ref. 39), *mail1-1* (GK-840E05) (ref. 24) and *mail3-2* (SALK_088600) (ref. 24) are all in the Columbia (Col-0) background. The *atxr5 atxr6* and *atmorc6-3*, and transgenic L5 lines were kindly provided by S. D. Michaels (Indiana University, Bloomington, IN, USA), S. E. Jacobsen (University of California, Los Angeles, CA, USA) and H. Vaucheret (INRA, Versailles, France), respectively. The *kun/mail1-2* mutation was genotyped by dCAPS (primers used in this study are listed in Supplementary Data 4). PCR products were digested with *Hpa*II restriction enzyme.

Plants were grown on soil or *in vitro* at 23 °C under a 16-h-light/8-h-dark cycle. For *in vitro* analysis, seeds were surface sterilized and sowed on solid 1/2 Murashige and Skoog medium containing 1% sucrose (w/v), supplemented with either 4 μM of DNA methylation inhibitor 5-aza-2′-deoxycytidine diluted in DMSO (A3656; Sigma-Aldrich) or with an equivalent concentration of DMSO for control plates.

**Mutagenesis and mapping.** About 25,000 seeds from the L5 line were mutagenized with 0.23% (V/V) ethyl methanesulfonate (EMS) (M0880; Sigma-Aldrich) for 14 h at room temperature. Following EMS neutralization, seeds were washed several times with water and subsequently planted on soil. Plants were allowed to self-fertilize, and seeds were collected in bulks of 150 plants. For each M1 pool, ∼1,500 M2 seeds were grown in soil for 2–3 weeks. To screen for mutants impaired in transcriptional gene silencing, one leaf per M2 plant was dissected, and leaves from four plants were together incubated at 37 °C for 24 h in 400 μg ml⁻¹ 5-bromo-4-chloro-3-indolyl-β-D-glucuronic acid, 10 mM EDTA, 50 mM sodium phosphate buffer pH 7.2, 0.2% triton X-100. To isolate mutant candidates, a second round of screening was applied to the corresponding M2 individuals in each M2 pool containing GUS-positive leaves.

We performed mapping-by-sequencing[40] using genomic DNA from a pool of 60 individual F2 mutant plants segregating from a *kun* x *Ler* cross. A library of short inserts was generated by breaking of the DNA into 50–500 bp fragments, ends repairs, 3′ A addition, ligation of adapters, and PCR amplification to generate the DNA colonies. The library was sequenced on a Illumina HiSeq 2500 instrument as 100 bp paired-end reads at Fasteris S.A. (Geneva, Switzerland). Reads were mapped on the *Arabidopsis* TAIR10 genome using BWA with the default options. Single-nucleotide polymorphisms calling was performed using samtools mpileup followed by bcftools view with the -bvgN option. Distribution of the obtained Col-0/*Ler* single-nucleotide polymorphisms was analysed using Next_Gen Mapping at http://bar.utoronto.ca/ngm/. Five candidate genes with EMS-induced non-synonymous mutations were identified in the mapping interval on chromosome 2. T-DNA mutant lines for the five candidate genes were analysed for release of gene silencing, allowing identification of *MAIL1*.

**Complementation of *kun*.** For the p35S::gMAIL1 and pMAIL1::gMAIL1 constructs, genomic fragments containing the *MAIL1* gene (+1 to +1,857 and −595 to +1,854, respectively, relative to the ATG codon whose A is designated +1) were PCR-amplified from Col-0 genomic DNA and cloned into the pDONR/ZEO vector (Invitrogen) by BP recombination. These fragments were introduced into pH2GW7 and pH7FWG,0 binary vectors (http://gateway.psb.ugent.be/search), respectively, by LR recombination. For complementation studies, *kun* plants were transformed by *Agrobacterium*-mediated transformation[41].

**RNA analyses.** Total RNA was extracted from immature inflorescences (stages 1–12) or aerial parts of three-week-old seedlings using guanidine hydrochloride (8 M guanidine hydrochloride, 20 mM MES (2-morpholinoethane-sulfonic acid), 20 mM EDTA and 50 mM β-mercaptoethanol at pH 7.0). For gel-based RT–PCR analyses, 0.05 μg of DNase-treated (RQ1 RNase-free DNase; Promega) total RNA was reverse transcribed and PCR-amplified using the One-Step RT-PCR kit (Qiagen) in a final volume of 10 μl. For RT-qPCR analyses,

0.2 μg of DNase-treated (RQ1 RNase-free DNase; Promega) total RNA was reverse transcribed using the PrimeScript RT reagent kit (Perfect real time) (TaKaRa) in a final volume of 10 μl, using a mix of oligo-dT and random hexanucleotides. One microlitre of cDNA was used for subsequent amplification using SYBR Premix Ex Taq II (Tli RnaseH Plus) (TaKaRa) and the Eco Real-Time system (Illumina) in a final reaction volume of 15 μl. Amplification of *ACTIN2* was used as a reference. Relative quantities were determined using the 'delta-delta method' formula $2^{-((Ct\ target\ sample - Ct\ actin2\ sample) - (Ct\ target\ calibrator - Ct\ actin2\ calibrator))}$. At least two biological replicates were analysed for each genotype. Original images of gels are shown in Supplementary Fig. 19.

**RNA sequencing.** Total RNA from 18-day-old seedlings was treated with the RQ1 RNase-free DNase (Promega), cleaned up with phenol-chloroform and precipitated with ethanol. Sequencing libraries were generated using the TruSeq RNA stranded protocol (Illumina) and sequenced on a Illumina HiSeq 2500 instrument as 50 bp single-end reads at Fasteris S.A. (Geneva, Switzerland). Two independent replicate libraries per genotype were generated and sequenced. Reads were mapped to the *Arabidopsis* TAIR10 genome using TopHat and Bowtie2 with the very sensitive option allowing up to two mismatches and only uniquely mapping reads were retained. Read statistics are listed in Supplementary Table 2. Expression of loci was determined by calculating reads per kilobase per million mapped reads (RPKM). *P*-values to detect differential expression were calculated by Fisher's Exact test. False discovery rate (FDR) correction was calculated using the Benjamini–Hochberg correction. Differentially expressed loci were defined by log2 RPKM mutant/WT > 2 (upregulated) or < −2 (downregulated) and a FDR < 0.01. Only loci defined as differentially expressed in both biological replicates were retained. Re-analysis of previously published RNA-seq datasets from leaf tissue of *mom1-2* (two replicates; GSM938356 and GSM938357), *atmorc6-3* (one replicate; GSM925646) and their corresponding WTs (GSM938341, GSM938342 and GSM925644) was performed as described above, with the exception that all loci defined as differentially expressed in the single *atmorc6-3* replicate were retained.

**Whole-genome bisulfite sequencing.** Genomic DNA was extracted from 18-day-old seedlings (DNeasy Plant Maxi Kit; Qiagen) and libraries were prepared from 100 ng genomic DNA and as previously described[42]. Two replicate libraries were generated for each genotype. In brief, we used the TruSeq Nano DNA sample prep kit (Illumina) according to the manufacturer's instructions; gDNA was sheared to 350 bp. After adapter ligation, we bisulfite (BS) treated the DNA fragments using the Epitect Plus Bisulfite Kit (Qiagen); we doubled the BS incubation to achieve better conversion. BS-treated libraries were amplified using the Kapa Hifi Hotstart Uracil+ polymerase (Kapa Biosystems). BS libraries were sequenced on an Illumina HiSeq2000 instrument as 101 bp paired-end reads. For image analysis we used Illumina RTA 1.13.48. Bisulfite-converted reads were processed and aligned following the procedure described in (ref. 42). In brief, the SHORE pipeline v0.9.0 (ref. 43) was used to trim and quality-filter the reads. Reads with more than 5 (or 2) bases in the first 25 (or 12) positions with a base quality score of below 5 were discarded. Reads were trimmed to the right-most occurrence of two adjacent bases with quality values of at least 5. Trimmed reads shorter than 40 bases were discarded. Reads were then aligned against the Col-0 reference genome sequence using GenomeMapper implemented in SHORE (ref. 40). Read statistics are listed in Supplementary Table 2.

**Identification of differentially methylated regions.** We first identified differentially methylated cytosines (DMCs) by smoothing the number of methylated and unmethylated reads per cytosine within each library using a Savitzky-Golay quadratic smoothing on 11 adjacent values. We used previously published data for *mom1-2*, *ddm1-2*, *atmorc6-3* and their corresponding WTs[19,27]. We then applied a Fisher's exact test (rounded smoothed #methylated reads/rounded smoothed #unmethylated reads) for each cytosine, comparing mutants and their corresponding WTs. Cytosines with a *P*-value ≤ 0.1 and an absolute methylation rate [smoothed #methylated reads/(smoothed #methylated reads + smoothed #unmethylated reads)] difference (mutant—WT) > 0.1 were deemed as DMCs. DMCs with, respectively, positive and negative methylation rate difference were called hyper-DMCs and hypo-DMCs. To define DMRs, DMCs of the same type (hypo or hyper) separated by a maximum distance of 160 bp (for CG), 240 bp (for CHG) or 70 bp (for CHH), were merged when no DMC of the other type was present within this distance. A Fisher's Exact test was applied to the rounded average number of smoothed methylated and unmethylated reads across the defined regions and resulting *P*-values were adjusted using the Benjamini–Hochberg correction (FDR). Regions with at least five DMCs, a minimum length of 50 bp, a FDR ≤ 0.05 and a methylation rate difference of at least 40% (for CGs) and 20% (for CHGs and CHHs) were deemed as DMRs. A DMR was considered mapping to a specific genomic feature when at least one nucleotide of the DMR overlapped with this feature.

**Small RNA sequencing.** Total RNA purified from immature inflorescences was used to generate small RNA libraries (TruSeq small RNA; Illumina), which were then sequenced on an Illumina HiSeq 2500 instrument (Fasteris S.A.). Reads were mapped to the *Arabidopsis* TAIR10 genome using TopHat without

mismatches. Only 21-nt and 24-nt uniquely mapping reads were retained. Read statistics are listed in Supplementary Table 2. For comparisons between libraries, read counts were normalized to the total amount of 18–26 nt mapping reads within each library.

**Small RNA gel blots.** Total RNA from immature inflorescences (15 μg) were heat-treated in 1.5 volume of standard formamide buffer (98% formamide, 10 mM EDTA), loaded on 15% polyacrylamide (19:1 acrylamide:bis-acrylamide)–8.3 M urea–0.5 × TBE gel and separated by electrophoresis. The samples were electroblotted to hybond-NX membranes (GE Healthcare) and fixed by carbodiimide-mediated crosslinking[44]. Pre-hybridization and hybridization were carried out in 5 × SSC, 20 mM Na$_2$HPO$_4$ pH 7.2, 7% SDS, 2 × Denhardt solution, 50 mg ml$^{-1}$ herring DNA at 50 °C. Membranes were washed twice with 3 × SSC, 25 mM NaH$_2$PO$_4$ pH 7.5, 5% SDS at 50 °C for 10 min, followed by one wash with 1 × SSC, 1% SDS at 50 °C for 10 min.

Hybridization signals were quantified using a phosphorimager (Molecular Imager FX; Bio-Rad). Original images of blots are shown in Supplementary Fig. 19.

**Chromatin immunoprecipitation.** ChIP were performed starting from 100–200 mg of 3–4-week-old seedlings using the low-cell ChIP kit (Diagenode), following manufacture's recommendations. Chromatin shearing was performed using the Bioruptor UCD-200 (Diagenode) for eight cycles of 15 s 'ON', 1 min 45 s 'OFF'. Immunoprecipitation was performed using antibodies to H3 (Abcam ab1791; 1 μg), H3K27me1 (Millipore #07-448, 2 μg for Supplementary Fig. 11c or Diagenode C15410045, 1.7 μg for Supplementary Fig. 13b) or H3K9me2 (Abcam ab1220; Diagenode C15410060; 2 μg). Four microlitre per 100 μl purified DNA was used for real-time PCR amplification using the Eco Real-Time system (Illumina) and SYBR Premix Ex Taq II (Tli RnaseH Plus) (TaKaRa) in a final reaction volume of 15 μl. Data were collected from three to five biological replicates, and relative enrichment was expressed as percentage of input.

**Immunocytology and DAPI-staining.** Rosette leaves from 4-week-old plants were fixed for 2 h in 4% formaldehyde in PBS and subsequently chopped on a glass slide. Tissues were covered with a coverslip, manually squashed and frozen in liquid nitrogen for 10 s before rapidly removing the coverslip. Slides were washed in PBS and pre-incubated with 3% BSA in PBS for 30 min at 37 °C. Slides were incubated overnight at 4 °C with primary antibodies diluted in 3% BSA in PBS (Diagenode anti-H3K9me2 (C15410060) diluted 1:100, Diagenode anti-H3K27me1 (C15410045) diluted 1:500 or Millipore anti-H3K27me1 (#07-448) diluted 1:100). Slides were washed in PBS and incubated with the secondary antibody (goat anti-rabbit Alexa 488, A-11008; ThermoFisher Scientific) for 1 h at 37 °C. After three washes in PBS, DNA was counterstained with DAPI in Vectashield mounting medium (Vector Laboratories). Nuclei were visualized using a Zeiss Axio Imager Z.1 epifluorescence microscope (Carl Zeiss AG) with a PL Apochromat × 100/1.40 oil objective. Z-stack images were captured with a Zeiss AxioCam MRm camera using the Zeiss ZEN software. The relative heterochromatin area was computed for each nucleus by calculating the ratio of the sum of chromocenter areas over that of the entire nucleus using the ImageJ software. To analyse nuclear localization of MAIL1 and MAIN, immunostaining was performed as described above using ProMAIL1:MAIL1-GFP (ref. 24) and ProMAIN:MAIN-GFP (ref. 39) transgenic lines (Col background) with anti-GFP antibody (MBL #598, 1:100 dilution).

**Fluorescent in situ hybridization.** Three-week-old seedlings were fixed in ethanol/acetic acid (3:1), washed twice in water and twice with citrate buffer at pH 4.5 (ref. 17). Tissues were digested for 3 h at 37 °C using cellulase, pectolyase and cytohelicase, all diluted in citrate buffer (0.3% w/v). Cells were spread onto a glass slide, incubated with 60% acetic acid for 1 min at 45 °C, and covered with ethanol/acetic acid (3:1). Slides were washed in water, post-fixed in 2% formaldehyde in PBS, washed in water and air-dried. Slides were then incubated 1 h in an ethanol/acetic acid solution (3:1) and air-dried. After a 30-min incubation at 60 °C, slides were treated with RNase A (100 μg ml$^{-1}$ in 2 × SSC) for 1 h at 37 °C, washed three times in 2 × SSC, fixed in 2% formaldehyde in PBS, washed twice in PBS, dehydrated in an ethanol series and air-dried. For hybridization, 1 μl of a PCR-amplified digoxigenin-11-dUTP probe for 106B repeats was diluted in 50% formamide, 2 × SSC, 50 mM sodium phosphate (pH 7), 10% Dextran sulphate. After adding the hybridization mixture, DNA was denatured at 80 °C for 2 min and slides were incubated overnight at 37 °C. Slides were successively washed at 42 °C in 2 × SSC, 0.1 × SSC and 2 × SSC. A last wash was performed at room temperature using 0.2% Tween-20 in 2 × SSC. Slides were pre-blocked with 5% non-fat dry milk in 4 × SSC, washed three times in 4 T (4 × SSC with 0.05% Tween-20) and once in TNT (100 mM Tris-HCl, pH 7.5, 150 mM NaCl, 0.05% Tween-20). For detection, slides were incubated with a mouse anti-digoxigenin antibody (1333062910; Roche) diluted at 1:125 in TNB (100 mM Tris-HCl, pH 7.5, 150 mM NaCl, 0.5% blocking reagent (Roche)). After three washes in TNT, slides were incubated with a rabbit anti-mouse antibody coupled to fluorescein isothiocyanate (F0257; Sigma) diluted at 1:500 in TNB. After three washes in TNT, slides were incubated with an Alexa 488-conjugated goat anti-rabbit antibody (A-11008; ThermoFisher Scientific) diluted at 1:100 in TNB. After three washes in TNT, slides were dehydrated in an ethanol series. DNA was counterstained with DAPI in Vectashield mounting medium (Vector Laboratories). Fluorescence images were captured using a Zeiss Axio Imager Z.1 epifluorescence microscope equipped with a Zeiss AxioCam MRm camera (Carl Zeiss AG).

**Trypan blue staining.** Isolated leaves were incubated for 1 h in staining solution (1 vol. 2.5 mg ml$^{-1}$ trypan blue in lactophenol, 2 vol. ethanol) after boiling. To detect cell death, stained material was incubated in chloral hydrate solution (2.5 g ml$^{-1}$ chloral hydrate in water) for 3 days (ref. 45).

**Flow cytometry.** Nuclei were prepared using the CyStain UV Precise P kit (Partec), according to the manufacturer's recommendations. Flow cytometry profiles were obtained on an Attune Acoustic Focusing Cytometer (Applied Biosystems) and analysed with the Attune Cytometric software (Applied Biosystems).

**Histone extraction and immunoblot.** Two grams of 4-week-old seedlings were ground in liquid nitrogen and transferred to 25 ml extraction buffer 1 (EB1: 0.4 M sucrose, 10 mM Tris-HCl pH 8, 10 mM MgCl$_2$, 5 mM β-mercaptoethanol, 0.1 mM phenylmethylsulfonyl fluoride and protease inhibitor tablet (Roche)). The suspension was filtered twice through a layer of Miracloth and centrifuged at 3,000g for 20 min at 4 °C. The pellet was resuspended in 300 μl EB2 (0.25 M sucrose, 10 mM Tris-HCl pH 8, 10 mM MgCl$_2$, 1% Triton X100, 5 mM β-mercaptoethanol, 0.1 mM phenylmethylsulfonyl fluoride, and protease inhibitor tablet (Roche)), then centrifuged at 16,000g for 10 min at 4 °C. The pellet was resuspended in 300 μl EB3 (1.7 M sucrose, 10 mM Tris-HCl pH 8, 2 mM MgCl$_2$, 0.15% Triton X100, 5 mM β-mercaptoethanol, 0.1 mM phenylmethylsulfonyl fluoride and protease inhibitor tablet (Roche)). Three hundred microlitres of EB3 was added to a new 1.5 ml tube and the resuspended pellet was carefully layered onto it. Following centrifugation at 16,000g for 60 min at 4 °C, the pellet was resuspended in HCl buffer (10 mM Tris-HCl pH 7.5, 5.2 mM EDTA, 0.25 N HCl, 5 mM β-mercaptoethanol and protease inhibitor tablet (Roche)), incubated 30 min on ice and then sonicated for four cycles of 20 s 'ON', 1 min 'OFF' (Bioruptor UCD-200; Diagenode) to shear chromatin. After centrifugation at 12,000g for 10 min at 4 °C, proteins were TCA-precipitated and resuspended in lane marker reducing sample buffer (Pierce). Approximately 1/10 (per lane) of the acid extraction was resolved on a 15% SDS-PAGE. After electroblotting on an Immobilon-P membrane (Millipore), western blot analysis was performed using anti-H3K27me1 (Diagenode, C15410045, 1:1,000), and anti-H4 (Abcam, ab10158, 1:2,000) antibodies, with BlockAce (Bio-Rad) as blocking reagent and WesternBright Sirius (Diagomics) as chemiluminescent substrate.

**Phylogenetic analysis.** Blast searches (blastp and tblastn) were performed starting from known *A. thaliana* PMD domains on the fifty-two species representing the diversity of the Viridiplantae lineage at the JGI Phytozome (V11) genomic resource (https://phytozome.jgi.doe.gov/pz/portal.html). Each time a new PMD was found in a given species, it was itself used as a query in a new BLAST search. To build the phylogenetic tree, we selected PMDs from 33 representative species (see Supplementary Table 1 for a list of species). All PMDs from PCGs from these 33 species were collected as well as representative versions of TE-associated PMDs when present. This led us to collect initially 464 proteins from the 33 species. We next performed a preliminary phylogenetic analysis of PMDs independently for each species. From these individual analyses, we were able to select PMDs that are truly representative of the diversity within each of our 33 species leading to the 308 proteins that were aligned and used to build our final phylogenic tree (Supplementary Data 3). Sequences were aligned using the multiple sequence comparison by log-expectation (MUSCLE v3.7) software[46]. Trees were reconstructed using the fast maximum likelihood tree estimation program PHYML[47] using the LG amino acids replacement matrix[48]. Statistical support for the major clusters were obtained using the approximate likelihood-ratio test (aLRT)[49]. To confirm the association of a PMD motif with a TE, we first used the Repbase database (http://www.girinst.org/) to identify PMD-containing Gypsy and PMD-containing MULE annotated in this database and present in our selected species. These sequences were used as references to identify putative new TE-associated PMDs in our phylogenic analysis. Each time a PMD clustered with at least one annotated TE in our phylogenic analysis, the genomic sequences surrounding the PMD were collected and translated (using Fgenesh available at http://www.softberry.com) to confirm TE association. To test the non-association with TEs of 'genic' PMD-B and PMD-C, the genomic region surrounding these motifs was submitted to CENSOR, (http://www.girinst.org/censor/index.php) the repeat masking tool of Repbase, to validate that these 'genic' PMDs were not embedded within a TE.

**Statistical analysis.** All statistical evaluations were performed under the R environment. Differences in mean for RT-qPCR and real-time PCR data were assessed using an unpaired, two-sided Student's *t*-test with Welch's correction. For relative chromocenter areas, Kruskal–Wallis rank sum test were performed with the native kruskal.test function, and Dwass–Steel–Critchlow-Fligner *post-hoc* tests were made using the pSDCFlig function with the asymptotic method from the

NSM3 package. The means and s.e.m. were derived from independent biological samples.

**Data availability.** Sequencing data generated in this study have been deposited to European Nucleotide Archive (ENA) under accession number PRJEB15202 (http://www.ebi.ac.uk/ena/data/view/PRJEB15202). All other data supporting the findings of this study are available within the manuscript and its Supplementary Files or are available from the corresponding author upon request.

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

## Acknowledgements

We thank L. López-González (Université Clermont Auvergne) and M. Fujii (Okayama University) for technical assistance, T. Hirayama (Okayama University) for experimental support and helpful discussions, and N. Suzuki (Okayama University) for support with confocal microscopy analysis. This work was supported by a grant from the European Research Council (ERC, I2ST 260742 to O.M.), an EMBO Young Investigator award (to O.M.), a Young Researcher grant from the Auvergne Regional Council (to O.M.), and the Max Planck Society and DFG (SFB 1101 to D.W.). Y.I. was supported by the Japan Society for the Promotion of Science (JSPS) postdoctoral fellowship for research abroad, a Grant-in-Aid for Research Activity Start-up (26891018) and for Young Scientists B (15K18578), and Program to Disseminate Tenure Tracking System, MEXT, Japan. P.B. was supported by a PhD studentship from the Ministère de l'éducation nationale, de l'enseignement supérieur et de la recherche.

## Author contributions

O.M. supervised the study. Y.I., T.P., J.-M.D. and O.M. conceived the study. Y.I., T.P., P.B., C.B. and M.-N.P.-P. conducted laboratory experiments. C.B. and D.W. contributed the whole-genome bisulfite sequencing data. M.W. created the *MAIL2*-RNAi line. R.P. and O.M. performed bioinformatics analyses. J.-M.D. conducted the phylogenetic analysis. Y.I., T.P., C.B., D.W., J.-M.D. and O.M. prepared the manuscript.

## Additional information

**Competing interests:** The authors declare no competing financial interests.

