## [Peer Review File · Nature Communications]

Reviewers' comments:

Reviewer #1 (Remarks to the Author):

The manuscript entitled Plants coopt a transposon-derived protein domain for gene silencing by Ikeda et al, describes identification of a novel gene silencing pathway in *Arabidopsis thaliana*.

First, the authors describe a mutant screen to identify novel hypermethylating factors, and identified the KUN locus which was subsequently NGS-mapped to MAIL1. This gene was confirmed with an independent T-DNA mutant previously described in Uehlken et al. 2014, as well as through a complementation assay with a transgene, where silencing of a GUS marker gene was restored.

To show that MAIL enforces gene silencing through yet unknown pathways, the authors tested and excluded the known major silencing pathways. Loss of MAIL1 results in upregulation especially in TE dense pericentromeric heterochromatin. Analysis of whole-genome bisulfite data led the authors to exclude DNA methylation as cause for upregulation. Introduction of mail1 into a hypermethylated ddm1 background enhanced transcript levels further, in contrast to in both mutant backgrounds alone. MAIL1 as part of the siRNA gene silencing pathway was also excluded using rdr2 mail1 double mutants and RNA blotting as well as small RNA sequencing. Comparisons with MOM1 and AtMORC1, which utilize two separate silencing pathways, also identified synergistic rather than complementary effects of MAIL1 based gene silencing. Further, the role of MAIL1 in physical chromatin condensation was shown cytogenetically. MAIL1 contains a plant mobile domain, and in order to analyze other members of this small gene family, RNAi lines identified loss of silencing and heterochromatin decondensation for MAIN, but not MAIL2 or MAIL3. Because main mail1 double mutants had no additional effects, the authors conclude that both function in the same pathway. Generally, the authors conclude that the plant mobile domain has a role in silencing, and subsequently identify, cluster and analyze PMD proteins from various plant species.

Concerns:

My major concern is with respect to the PMD protein analysis and final conclusions. The study is well designed and described up to the point of PMD protein analysis. This last part feels disconnected from the main discovery. It is very speculative, and important details are outsourced into the figure legend. Please provide background information on PMDs (structure, function, ...). There is not a single literature reference in this whole paragraph. If the PMD-A clade is ancestral, would the authors not expect some in *Arabidopsis* too? Are they present in "older" Brassicaceae? Are the TE-PMD fusions correctly annotated in *Malus*, *Sorghum* etc.? There is also a recent example for transposase derived proteins with silencing function by Li et al. [10.15252/embj.201694284](https://doi.org/10.15252/embj.201694284)

The conclusion should focus on *Arabidopsis*, because the authors provide evidence for a novel silencing pathway only for *A. thaliana*. In fact, the paper does not show a single PMD and TE-fusion in *A. thaliana*. I further disagree that this study "demonstrates" that MAIL1 and MAIN "evolved from a TE encoded pathway". Actual evidence, or a literature link is missing here.

While the first part of the manuscript is very descriptive, the authors should also expand on

how the chromatin condensation could be initiated by a MAIL1/MAIN dimer. Can these proteins not do it individually? Anything in the literature?

The Methods section requires some minor work. Company names are used inconsistently, e.g. w/ or w/o city and state names. Reagent kits are not always correctly labeled.

Reviewer #2 (Remarks to the Author):

Ikeda et al show that the loss of MAIL1, a plant mobile domain protein, results in increased TE expression, mostly from the pericentric regions. Based on additive/synergistic effects on expression or chromatin modification when mail1 is combined with mutations in ddm1, rdr2, mom1, atmr6 or atxr mutants, Ikeda et al conclude that MAIL1 silences loci independently of DNA methylation and small RNAs. These observations led to the conclusion that MAIL1 acts outside of previously known pathways to repress TEs. Based on the data and analyses presented this conclusion is premature. Additionally, the major claim in the final sentence of the abstract, that MAIL1 is a TE-derived protein coopted for silencing, which was then captured by TEs to promote its own interests, is insufficiently supported.

1. Ikeda et al profile DNA methylation genome-wide in WT and mail1 mutants and conclude there is no change in DNA methylation. The metaplots presented in Figure 2 and Extended Data Figure 3 do not provide the resolution needed to conclude that there is no methylation change at any individual locus. The authors should perform an analysis for differentially methylated regions between the wild type and mutant.

The authors also argue that the synergistic effects on expression observed in double mutant combinations between mail1 and ddm1 or mail1 and rdr2 indicate that mail1 does not alter DNA methylation. Because multiple DNA methylation pathways can target loci simultaneously, this argument is not correct. For example, in Extended Data Figure 5, qPCR analysis shows that SDC expression is increased in mail1 and rdr2 compared to wild type, but a much stronger increase is observed in mail1 rdr2 double mutants. The same expression effect would be observed in rdr2 cmt3 double mutants because both RdDM and CMT3 methylation pathways silence SDC.

2. From Fig3A the authors conclude that mail1 chromocenters are larger than wild type and thus decondensed. This is not the definition of decondensed chromocenters generally used in the field. Chromocenters are defined as densely staining regions of the nucleus, and decondensed chromocenters would be indicated by reduced DAPI staining (as seen for ddm1 mutants in Extended Data Figure 8). These chromocenters look perfectly fine, but larger. It appears from the images shown that mail1 nuclei may also be slightly larger than wild type. Could chromocenters have larger areas purely as a result of having larger nuclei? MAIL1 nuclei could be larger if they have increased ploidy. Mutants in some atxr genes are known to have increased replication of the pericentric regions, which could be detected in mail1 mutants by flow cytometry. Finally, if the authors believe there is a mail1 chromocenter phenotype, they should assess whether the MAIL1 protein is localized to chromocenters.

3. RNA-seq and small RNA-seq experiments were carried out in different plant tissues. This

is problematic for Figure 2D, from which the conclusion is drawn that siRNAs accumulation is not altered for mail1 upregulated TEs. This comparison is best carried out in the same tissue. Additionally, how was the small RNA sequencing data normalized between mutants and wild-type?

4. The phylogeny in Figure 3D is the only data presented in support of the authors claim for exaptation and subsequent recapture of MAIL1 and related proteins from/by TEs. The following questions/experiments arise: 1) Alignment between a gypsy TE PMD and MAIL1 and a MULE TE PMD and MAIL1 should be shown. 2) If the ancestral state of PMDs is as part of a Gypsy TE, why is this configuration only observed in 4 species? The Gypsy TE PMD is not found in the basal angiosperm *Amborella*. 3) Are the PMD-A Gypsy TEs related outside of the PMD domain? 4) At minimum one would need to show the production of a PMD-MULE fusion transcript in order to even think about concluding it was "serving" (line 148) the TE. There is no evidence presented that "...this domain can be reintegrated by TEs to once again serve their function" (line 30).

5. Experiments from multiple species show that TEs can be upregulated because of mutations in genes that function in a wide range of pathways. These pathways may be centered on TE defense (such as the RdDM pathway), or other cellular pathways involved in transcript processing, such as the nuclear exosome. Additionally, TEs may be upregulated under stress conditions or in response to other phenomena. Thus the authors may want to consider the following as they work to understand the function of MAIL1:

1) Cell death has been previously suggested to lead to increased TE expression, potentially even through a cell non-autonomous pathway (Xie et al, *Genetics* 188, 2011). Can the cell death seen in MAIL1 vascular tissues (Uhlken et al, *Plant J*, 2014) account for increased TE expression?

2) The nuclear exosome has been previously implicated in the repression of TE expression (Shin et al, *Plos Genetics*, 2013). The authors may want to test expression in mail1-1; rrp4 and mail1-1; rrp41 mutants.

Reviewer #3 (Remarks to the Author):

In the current submission, Ikeda et al. describe the results of a genetic screen for factors that release the silencing of the L5 GUS reporter. In addition to known silencing factors, they identified a new gene, MAIL1, that is required silencing of the L5 reporter and several other endogenous targets. The authors go on to carry out a thorough genetic characterization of this mutant relative to known silencing pathways. These analyses clearly demonstrate a role for MAIL1 that is independent of both DNA methylation and siRNAs, placing it amongst a very small group of factors acting that influence gene silencing independently of DNA methylation. Given the importance of transposon silencing and the limited understanding of events occurring downstream/independently of DNA methylation, these findings represents a significant advance and will be of broad interest to the fields of chromatin and gene regulation. However, enthusiasm for the results is slightly dampened due to weaknesses in the reporting of the global analyses and weak effects of the mail1

mutant on chromocenter condensation. These areas could be significantly strengthened by addressing the issues detailed below.

1. Given the effects of the *mail1* mutation on H3H27me1 levels, a more detailed analysis of the genetic connections between *mail1* and *axr5/6*, two H3k27 mono-methyltransferases, should be conducted to more definitively show these factors are functioning in independent pathways. qRT-PCR analyses similar to those presented in Fig 2b should be sufficient.

2. To fully evaluate the authors' conclusions about the global effects of the *mail1* mutation, additional information is required. For all NGS data, general statistics on the read depth, mapping stats, conversion rates etc should be presented.

Of particular relevance to the authors claims are genes upregulated in the *mail1* mutants, yet no gene list are presented. Furthermore, the presentation of the expression data via metagene analyses seems misleading as it suggests a massive, and global, upregulation of TE etc. in pericentromeric heterochromatin yet according to extended data figure 3a only around 350 genes are upregulated genome-wide. Without knowing the relative coverage of the datasets being compared and the fraction of differentially expressed genes shown in extended data figure 3a that fall into pericentromeric heterochromatin it is not possible to evaluate the authors claim that *mail1* mutants selectively affect the silencing of pericentromeric loci.

3. Like the data linking the *mail1* mutation to the release of silencing at pericentromeric heterochromatin, the data linking this mutation to chromocenter decompensation is also weak. Although the authors quantify the differences, it is not easy to disentangle the differences in nuclei and chromocenter size, especially considering the subtleness of the effect in comparison to other mutants, like *ddm1*, that show a strong chromocenter decondensation phenotypes. Luckily, the authors show increased in expression of the 106B pericentromeric repeats in the *mail1* mutation. Thus, the addition of a FISH experiment using a probe to this repeat may allow for a more robust connection between chromocenter compaction and gene expression.

4. The logic regarding the phylogenetic claims appear rational, however, this reviewer defers to other reviewers with more expertise in the area to fully evaluate the authors claims.

Reviewer #4 (Remarks to the Author):

The manuscript by Mathieu et al. evidences a new pathway for gene(TE)-silencing in plants, which seems to represent a great advance in the plant research domain. I trust the works done being quite solid, based on the genetic and NGS analyses, but I admit I'm not competent in plants genetics, and then not competent to foresee the impact of the paper in the future.

As a general comment, I'm surprised by the shortness of the manuscript (about 1500 words for main text and 1700 for methods, and 3 figures only, while, if I'm not wrong, Nature Communications allows up to 5000 words in the main text and 3000 in the methods, and up

to 10 figures. It is for sure that the huge amount of work done would deserve a more developed manuscript, and I would advice the authors to profit of this opportunity to better detail their work. For now it sounds like a big summary, which leaves the reader a bit frustrated...

The part about evolution, I have been asked to review, would of course benefits from this manuscript expansion. Yet I have the feeling it is not the main groundbreaking point of the manuscript (although the cooptation notion is in the title, in this way, the title may not be adapted unless some more work is done on the evolution of the domain). In my opinion, efforts should be kept for expanding a bit the genetic part. Anyway I believe the last part on domain evolution could be improved by adding some more methodological details, and some other modifications. If the title has to be kept, a real effort would be necessary to reinforce this part of the manuscript, which for now, appears like a very small conclusive paragraph. In particular, the hypothesis of an exaptation by the genome of a TE protein (as opposed to the alternative hypothesis of a capture of genomic sequence by transposons) should be better supported by a more in-depth analysis, if possible.

Below are my specific comments on this part:

Point 1:

I don't see any argument to state that the PMD-A clade (gypsy) is ancestral. In absence of any outgroup, a phylogenetic tree cannot be rooted, and the PhyML proposed topology is totally arbitrarily, unless you specify which of the sequences are outgroup (in some phyML versions, specification is done by adding a * to the sequence name.). I'm not sure this has been done, and if so, I would expect some arguments for choosing PMD-A as the outgroup, rather than another one. In any other case, the tree should be kept as unrooted. This point is important since it seems to be the initial statement on which the further evolutionary scenario is based.

Point 2:

I regret that, while obvious efforts have been done to choose representative (and "basal") species of dicot and monocot, the tree in the figure does not highlight the species distribution. I would suggest adding some indications about at least the origin (angiosperm, monocot or dicot) of each sequence (by marks or colors...). Discussion about this could also be interesting, since it can give indication about the time at which suspected events (exaptation, recapture...) took place. (Here I suspect it may be difficult, but it is worth to try.)

Point 3:

It seems clear that few efforts have been done previously to characterize the PMD domain (Babu et al. (2006) actually focused on another domain, and the PMD description is just a by-product of their analysis). However, there is another reference dealing a bit with this PMD domain in plants gypsy-like elements (Steinbauerova et al 2011 Genetica) that could be cited.

Point4:

Furthermore, although not that easy to find, the PMD domain is already defined in the pfam database (PF10536), based on similarities found on several thousands of sequences

(<http://pfam.xfam.org/family/PF10536#tabview=tab3>). A full set of 2700 aligned sequences, or a representative set of 65 sequences are available for downloading. These sequences come from the uniprot database, and have been identified through a HMM-based strategy. I think it is a great idea to identify this domain directly from the genome DNA sequences, even if the BLAST (TBLASTN) methodology is probably less efficient than HMM. However, by restricting the search to the 52 genomes of the phytozome, or to the 9 selected species, some interesting data may be lost. This huge amount of data in the pfam database should be exploited in order to gain support to the proposed scenario. I would suggest the authors to explore this huge dataset, to refine their statement as well as to confirm their phylogenetic analysis. From what I tried (raw phylogenetic analysis of the 2700 sequences, and phylogenetic analysis of the 65 representatives sequences + the authors sequences), the phylogeny of the authors is of course a simplification but globally stay in agreement with the tree obtained with the full set of sequences. Again, this dataset would provide great materials for a deeper/wider analysis of the domain.

Point 5:

The authors propose a scenario in which the PMD domain was first present in Gypsy elements, then has been exapted by genomes and has diverged into two groups, before some sequences from one group being captured by MULEs elements. Noticeably, this scenario is the opposite of the one proposed in Steinbauerova et al 2011 *Genetica*, who suggest the PMD domain was transmitted to gypsy elements after insertion of a MULE element within a gypsy element.

I have to say that, based on the arguments presented here and there, I'm not fully convinced by any of these scenarios (although one of them should of course approach the truth...). I found anyway an interesting remark in Steinbauerova et al 2011: The identified PMD-carrying Gypsy elements (Tat group) are relatively recent elements (identified by LTR-Harvest if I remember well), meaning ancient elements are not identified and then cannot be used to date the insertion event. In the present manuscript, the phylogenetic analysis, compared to species phylogeny, could bring some useful information. The fact that only recent Gypsy elements are identified should also be taken into account for the discussion.

Point 6:

It is said that the PMD-A is exclusively found in Gypsy elements. There is no detail in the method part of how the presence of gypsy sequences on both sides of the PMD-A domains have been verified. From what I saw, rice sequences are annotated as Gypsy, but I could not find annotation for the other species. Of course it is probably because I'm not familiar with phytozome website. However I think it would be valuable to mention whether the information is drawn from annotation files for the species, or from independent checking of the flanking sequences. My underlying question is whether this clade is really exclusively found in Gypsy elements (Tat group?). I suppose so for Os and Md sequences, based on annotation or on the fact they are highly repeated in the genome. Yet, I'm not even sure for Zm9 and Sb sequences (Curiously, for Zm9, I was even unable to retrieve the real sequence on the *Zea mays* genome at the phytozome or at NCBI!). Here also, checking on the data provided by the pfam database may help to support the assertion that PMD-A is exclusively made of gypsy sequences.

The same remarks stand for the two other clades, and the presence or absence of TE-carried PMD elements, in particular, the absence of any TE among the PMD -B clade. Again, pfam data would be useful, along with own verifications about the fact that sequences are

part of TEs or not, within each clade.

Point 7:

I will admit that the details I'm asking for will never be sufficient to be sure about the proposed scenario. Besides the data provided, beside the improvement that could be brought, there is also the big problem of horizontal transfers, frequent for TEs, that can blur the results, making any scenarios always speculative, unless hardcore TE phylogeny analysis is done in parallel... (yet I don't think it is the point here).

This does not mean that the scenario is not plausible, but just that the authors should be more cautious when stating that "this study demonstrates that MAIL1 and MAIN,...., and that evolved from a TE -encoded pathway via domestication of a TE protein.".

Point 8:

Finally, also concerning the conclusion, although this time, the authors precise that it is a speculation, I would like to pinpoint the fact that the presence of some protein domains within TEs is not at all an evidence of a functional role, nor the fact that this domain is frequently found in TEs (since non functional sequences can just be amplified as a consequence of TE amplification). Here it is obvious that the functional role (within TE) is far from elucidated, although the work on MAIN and MAIL1 give some important insights on the role this domain may have.

I acknowledge that my recommendations imply some substantial work, and there is no warranty that any better-supported evolutionary scenario may emerge. Yet I believe it would greatly improve this part of the manuscript, and hopefully gives more support to the proposed title of the manuscript.

Reviewers' comments:

Reviewer #1 (Remarks to the Author):

The manuscript entitled Plants coopt a transposon-derived protein domain for gene silencing by Ikeda et al, describes identification of a novel gene silencing pathway in *Arabidopsis thaliana*. First, the authors describe a mutant screen to identify novel hypermethylating factors, and identified the KUN locus which was subsequently NGS-mapped to MAIL1. This gene was confirmed with an independent T-DNA mutant previously described in Uehlken et al. 2014, as well as through a complementation assay with a transgene, where silencing of a GUS marker gene was restored. To show that MAIL enforces gene silencing through yet unknown pathways, the authors tested and excluded the known major silencing pathways. Loss of MAIL1 results in upregulation especially in TE dense pericentromeric heterochromatin. Analysis of whole-genome bisulfite data led the authors to exclude DNA methylation as cause for upregulation. Introduction of mail1 into a hypermethylated ddm1 background enhanced transcript levels further, in contrast to in both mutant backgrounds alone. MAIL1 as part of the siRNA gene silencing pathway was also excluded using rdr2 mail1 double mutants and RNA blotting as well as small RNA sequencing. Comparisons with MOM1 and AtMORC1, which utilize two separate silencing pathways, also identified synergistic rather than complementary effects of MAIL1 based gene silencing. Further, the role of MAIL1 in physical chromatin condensation was shown cytogenetically. MAIL1 contains a plant mobile domain, and in order to analyze other members of this small gene family, RNAi lines identified loss of silencing and heterochromatin decondensation for MAIN, but not MAIL2 or MAIL3. Because main mail1 double mutants had no additional effects, the authors conclude that both function in the same pathway. Generally, the authors conclude that the plant mobile domain has a role in silencing, and subsequently identify, cluster and analyze PMD proteins from various plant species.

Concerns:

My major concern is with respect to the PMD protein analysis and final conclusions. The study is well designed and described up to the point of PMD protein analysis. This last part feels disconnected from the main discovery. It is very speculative, and important details are outsourced into the figure legend. Please provide background information on PMDs (structure, function, ...). There is not a single literature reference in this whole paragraph. If the PMD-A clade is ancestral, would the authors not expect some in *Arabidopsis* too? Are they present in "older" Brassicaceae?

The PMD has been very poorly characterized. In the revised manuscript, we refer to the only two papers that mention this domain (Babu et al 2006, and Steinbauerova et al 2011).
We have significantly extended our phylogenetic analysis of PMD domains in the revised manuscript. Please refer to our response to the point 1 of reviewer #4 for these questions.

Are the TE-PMD fusions correctly annotated in *Malus*, *Sorghum* etc.?

Please see our answer to reviewer #4, point 6 for these questions.

There is also a recent example for transposase derived proteins with silencing function by Li et al. [10.15252/embj.201694284](https://doi.org/10.15252/embj.201694284)

This paper by Li et al. has unfortunately been retracted (see EMBO J. 2016, 35, 2060).

The conclusion should focus on *Arabidopsis*, because the authors provide evidence for a novel silencing pathway only for *A. thaliana*. In fact, the paper does not show a single PMD and TE-fusion in *A. thaliana*.

Following the reviewer's recommendation, we have modified the title of the manuscript to "Arabidopsis proteins with a transposon-related domain act in gene silencing".

I further disagree that this study "demonstrates" that MAIL1 and MAIN "evolved from a TE encoded pathway". Actual evidence, or a literature link is missing here.

This statement has been removed from the revised manuscript.

While the first part of the manuscript is very descriptive, the authors should also expand on how the chromatin condensation could be initiated by a MAIL1/MAIN dimer. Can these proteins not do it individually? Anything in the literature?

Our genetic analysis suggests that MAIL1 and MAIN function in the same molecular pathway, potentially as a heterodimer, for both chromatin compaction and gene silencing. *mail1* and *main* single mutants exhibit similar defects in chromatin condensation than *mail1 main* double mutants indicating that MAIL1 and MAIN proteins are individually not sufficient for chromatin condensation. Unfortunately, the literature does not provide relevant insights, and a deeper understanding of the molecular mechanisms involved will have to await the identification and characterization of potential MAIL1/MAIN interacting protein partners.

The Methods section requires some minor work. Company names are used inconsistently, e.g. w/ or w/o city and state names. Reagent kits are not always correctly labeled.

The Methods section has been carefully edited for consistency and correctness in the revised version of the manuscript.

Reviewer #2 (Remarks to the Author):

Ikeda et al show that the loss of MAIL1, a plant mobile domain protein, results in increased TE expression, mostly from the pericentric regions. Based on additive/synergistic affects on expression or chromatin modification when *mail1* is combined with mutations in *ddm1*, *rdr2*, *mom1*, *atmorc6* or *atxr* mutants, Ikeda et al conclude that MAIL1 silences loci independently of DNA methylation and small RNAs. These observations led to the conclusion that MAIL1 acts outside of previously known pathways to repress TEs. Based on the data and analyses presented this conclusion is premature. Additionally, the major claim in the final sentence of the abstract, that MAIL1 is a TE-derived protein coopted for silencing, which was then captured by TEs to promote its own interests, is insufficiently supported.

This sentence has been modified; please refer to our response to the points 1 and 8 of reviewer #4.

1. Ikeda et al profile DNA methylation genome-wide in WT and *mail1* mutants and conclude there is no change in DNA methylation. The metaplots presented in Figure 2 and Extended Data Figure 3 do not provide the resolution needed to conclude that there is no methylation change at any individual locus. The authors should perform an analysis for differentially methylated regions between the wild type and mutant.

Following the reviewer's request, we have determined DMRs in *mail1* relative to the WT. For comparison purposes and as controls, we have also applied our DMR calling pipeline to previously published BS-seq datasets of *mom1*, *atmorc6* and *ddm1* mutants and their corresponding WTs. This analysis support our previous conclusion by revealing that there are few DMRs in *mail1* and that there is no correlation between transcriptional upregulation in *mail1* and reduced DNA methylation. This is presented in the new Figure 3b and supplementary Figure 6 of the revised manuscript.

The authors also argue that the synergistic affects on expression observed in double mutant combinations between *mail1* and *ddm1* or *mail1* and *rdr2* indicate that *mail1* does not alter DNA methylation. Because multiple DNA methylation pathways can target loci simultaneously, this argument is not correct. For example, in Extended Data Figure 5, qPCR analysis shows that SDC expression is increased in *mail1* and *rdr2* compared to wild type, but a much stronger increase is observed in *mail1 rdr2* double mutants. The same expression effect would be observed in *rdr2 cmt3* double mutants because both RdDM and CMT3 methylation pathways silence SDC.

It is certainly correct that multiple DNA methylation pathways can target loci simultaneously. However, we did not use data from mutant combinations to conclude about potential DNA methylation changes in *mail1*. We based our conclusion that DNA methylation is largely unaltered in

mail1 solely on the BS-seq data analysis of the single mutant. The analysis of the interaction between *mail1* and *ddm1* (or *mail1* and *rdr2*) mutations was only performed to determine whether MAIL1 controls silencing downstream or independently of DNA methylation (or siRNA production in the case of *mail1 rdr2*). Because some of the MAIL1-targeted loci we analyzed totally lose DNA methylation in *ddm1*, the synergy between *mail1* and *ddm1* mutations for silencing release strongly suggest that MAIL1 acts independently, rather than downstream, of DNA methylation. We have rewritten this part to better convey our message.

2. From Fig3A the authors conclude that *mail1* chromocenters are larger than wild type and thus decondensed. This is not the definition of decondensed chromocenters generally used in the field. Chromocenters are defined as densely staining regions of the nucleus, and decondensed chromocenters would be indicated by reduced DAPI staining (as seen for *ddm1* mutants in Extended Data Figure 8). These chromocenters look perfectly fine, but larger. It appears from the images shown that *mail1* nuclei may also be slightly larger than wild type. Could chromocenters have larger areas purely as a result of having larger nuclei? MAIL1 nuclei could be larger if they have increased ploidy. Mutants in some *atxr* genes are known to have increased replication of the pericentric regions, which could be detected in *mail1* mutants by flow cytometry. Finally, if the authors believe there is a *mail1* chromocenter phenotype, they should assess whether the MAIL1 protein is localized to chromocenters.

Please refer to our common response to this point and point 3 of reviewer #3 below.

3. RNA-seq and small RNA-seq experiments were carried out in different plant tissues. This is problematic for Figure 2D, from which the conclusion is drawn that siRNAs accumulation is not altered for *mail1* upregulated TEs. This comparison is best carried out in the same tissue. Additionally, how was the small RNA sequencing data normalized between mutants and wild-type? We carried out small RNA-seq (and RNA gel blots) on floral tissue because small RNAs are most abundant in this tissue and because we wanted to explore any potential role for MAIL1 in the accumulation of various types of small RNAs, including floral-specific PolIV-dependent siRNAs (type I) (Mosher et al., 2009, Nature, 460, 283). We have performed additional quantitative RT-PCR analyses to verify that representative loci upregulated in *mail1* seedlings are also upregulated in *mail1* floral tissue (see new Supplementary Figure 9). The sRNA-seq libraries were normalized to the number of 18-26nt uniquely mapped reads in each library (see new Supplementary Table 6); this is now mentioned in the Methods section of the revised manuscript.

4. The phylogeny in Figure 3D is the only data presented in support of the authors claim for exaptation and subsequent recapture of MAIL1 and related proteins from/by TEs. The following questions/experiments arise:

1) Alignment between a gypsy TE PMD and MAIL1 and a MULE TE PMD and MAIL1 should be shown.

The alignment is now shown in the revised manuscript (new supplementary figure 17). This alignment was made using two representative Gypsy TE PMDs, two representative MULE TE PMDs and MAIL1 PMD. Of course, MAIL1 PMD is much closer to the MULE TE PMDs since both belong to the same evolutionary clade (PMD-C).

2) If the ancestral state of PMDs is as part of a Gypsy TE, why is this configuration only observed in 4 species? The Gypsy TE PMD is not found in the basal angiosperm *Amborella*. Our new analysis of 33 species (instead of 9) reveals a large distribution of the PMD-A Gypsy in monocot and dicot species including *Amborella* (PMD-A Gypsy elements are present in 16 of the 33 species used). To explain this large distribution, we propose that the PMD-A Gypsy association took place in the common ancestor of angiosperm. The fact that PMD-A Gypsy elements are not found in all angiosperm species is not contradictory to this proposition. Indeed, the evolutionary success of a given transposable element family can vary from one lineage to another. In our case, PMD-A-

containing gypsy elements managed to amplify successfully (and recently enough to be observable) in the *Amborella* lineage, in the grass lineage as well as in the *Fabidae*, the *Malpighiales*, the *Solanales* and the *Vitales* lineages. However, with the exception of the *Theobroma cacao* species, it became extinct in the *Brassicales* lineages and therefore is not found in *Arabidopsis*.

3) Are the PMD-A Gypsy TEs related outside of the PMD domain?

Yes, they all belong to the Gypsy clade and share a Gypsy-like GAG-POL region. The difference with other Gypsy elements is that, in PMD-A Gypsy, the ENV region was replaced by a PMD-A motif coded in the opposite strand compared to GAG-POL.

4) At minimum one would need to show the production of a PMD-MULE fusion transcript in order to even think about concluding it was “serving” (line 148) the TE. There is no evidence presented that “..this domain can be reintegrated by TEs to once again serve their function” (line 30). Although we believe that our data strongly support that PMDs have been recaptured by DNA TEs twice independently during evolution, we fully acknowledge that further studies are needed to assess whether TE-associated PMDs play a functional role in TEs life cycle. In the revised manuscript, we have remove this statement that remains purely speculative at this point, and simply mention it as an attractive possibility at the end of the discussion section.

5. Experiments from multiple species show that TEs can be upregulated because of mutations in genes that function in a wide range of pathways. These pathways may be centered on TE defense (such as the RdDM pathway), or other cellular pathways involved in transcript processing, such as the nuclear exosome. Additionally, TEs may be upregulated under stress conditions or in response to other phenomena. Thus the authors may want to consider the following as they work to understand the function of MAIL1:

1) Cell death has been previously suggested to lead to increased TE expression, potentially even through a cell non-autonomous pathway (Xie et al, Genetics 188, 2011). Can the cell death seen in MAIL1 vascular tissues (Uhlken et al, Plant J, 2014) account for increased TE expression? Although *mail1-1* null mutants display readily observable cell death in mature tissues, we found that this is not the case in *mail1-2* plants (see new supplementary figure 3a). In addition, trypan blue staining experiments indicate that neither *mail1-2* nor *mail1-1* exhibit cell death in young tissues at the developmental stage used for RNA-seq analyses (see new supplementary figure 3b). Therefore, we do not believe that cell death accounts for loss of TE silencing in *mail1* mutants.

2) The nuclear exosome has been previously implicated in the repression of TE expression (Shin et al, Plos Genetics, 2013). The authors may want to test expression in *mail1-1; rrp4* and *mail1-1; rrp41* mutants.

This is an interesting suggestion we may address in future work, although we note that loss of silencing in exosome mutants is associated with a significant reduction in H3K9me2, which is not the case in *mail1* and *main* mutants. However, due to the necessity of breeding for new mutant combinations, such analysis would significantly delay the current submission.

Reviewer #3 (Remarks to the Author):

In the current submission, Ikeda et al. describe the results of a genetic screen for factors that release the silencing of the L5 GUS reporter. In addition to known silencing factors, they identified a new gene, MAIL1, that is required silencing of the L5 reporter and several other endogenous targets. The authors go on to carry out a thorough genetic characterization of this mutant relative to known silencing pathways. These analyses clearly demonstrate a role for MAIL1 that is independent of both DNA methylation and siRNAs, placing it amongst a very small group of factors acting that influence

gene silencing independently of DNA methylation. Given the importance of transposon silencing and the limited understanding of events occurring downstream/independently of DNA methylation, these findings represent a significant advance and will be of broad interest to the fields of chromatin and gene regulation. However, enthusiasm for the results is slightly dampened due to weaknesses in the reporting of the global analyses and weak effects of the *mail1* mutant on chromocenter condensation. These areas could be significantly strengthened by addressing the issues detailed below.

1. Given the effects of the *mail1* mutation on H3H27me1 levels, a more detailed analysis of the genetic connections between *mail1* and *atxr5/6*, two H3k27 mono-methyltransferases, should be conducted to more definitively show these factors are functioning in independent pathways. qRT-PCR analyses similar to those presented in Fig 2b should be sufficient.

We analyzed the genetic connection between *mail1* and *atxr5/6* by profiling and comparing transcriptomes of *mail1*, *atxr5/6* and *mail1 atxr5/6* triple mutants (see supplementary figure 13) by RNA-seq, which we believe provides the highest resolution. The high number of TEs specifically reactivated in *mail1 atxr5/6* strongly suggests that *MAIL1* and *ATXR5/6* function in largely distinct silencing pathways.

In addition, we now show that the heterochromatin over-replication phenotype of *atxr5/6* mutants is not observed in *mail1* nuclei (see supplementary figure 14e).

2. To fully evaluate the authors' conclusions about the global effects of the *mail1* mutation, additional information is required. For all NGS data, general statistics on the read depth, mapping stats, conversion rates etc should be presented.

Of particular relevance to the authors' claims are genes upregulated in the *mail1* mutants, yet no gene list is presented.

We now include summary statistics for all NGS data (see supplementary Table 6) as well as lists of transposons and protein-coding genes misregulated in the various mutants analyzed (see supplementary Tables 1 and 2).

Furthermore, the presentation of the expression data via metagene analyses seems misleading as it suggests a massive, and global, upregulation of TE etc. in pericentromeric heterochromatin yet according to extended data figure 3a only around 350 genes are upregulated genome-wide. Without knowing the relative coverage of the datasets being compared and the fraction of differentially expressed genes shown in extended data figure 3a that fall into pericentromeric heterochromatin it is not possible to evaluate the authors' claim that *mail1* mutants selectively affect the silencing of pericentromeric loci.

We did not intend to convey the message that "mail1 mutants selectively affect the silencing of pericentromeric loci" (and we must respectfully point out that we did not make such claim). We included the metagene representation to illustrate that *mail1* mutation mostly leads to transcript upregulation, including at loci located within pericentromeric heterochromatin. As requested, we now provide lists for loci upregulated in *mail1* and other mutants (supplementary table 1). In addition, we now show the relative proportion of *mail1*-upregulated loci located in euchromatin and in heterochromatin (new figure 2a,b). This shows that *mail1*-upregulated TEs are significantly enriched for TEs localized in pericentromeric heterochromatin, while upregulated protein-coding genes do not show such enrichment. Therefore, *MAIL1* controls silencing of heterochromatic TEs as well as expression of euchromatic genes. The corresponding paragraph has been modified to better convey this conclusion.

3. Like the data linking the *mail1* mutation to the release of silencing at pericentromeric heterochromatin, the data linking this mutation to chromocenter decompensation is also weak. Although the authors quantify the differences, it is not easy to disentangle the differences in nuclei and chromocenter size, especially considering the subtlety of the effect in comparison to other

mutants, like *ddm1*, that show a strong chromocenter decondensation phenotypes. Luckily, the authors show increased in expression of the 106B pericentromeric repeats in the *mail1* mutation. Thus, the addition of a FISH experiment using a probe to this repeat may allow for a more robust connection between chromocenter compaction and gene expression.

Following is our common response to this point and to point 2 raised by reviewer #2:

Our measurements of chromocenters area are normalized to the area of the whole nucleus; therefore, they are indicative of changes in chromocenter size, independent of changes in nuclear size. Nonetheless, nuclear size might also be altered in *mail1* mutants. To address this possibility, we have carefully measured nuclear area of 3-week-old WT and mutant seedlings and found that nuclear size is not significantly modified in *mail1* mutants (see new supplementary Figure 14c). In addition, we have performed flow cytometry analysis of *mail1*, *main*, *mail1 main* and *atxr5/6* mutants as suggested by reviewer #2. This analysis revealed that nuclei of *mail1*, *main* and *mail1/main* do not have increased ploidy and do not display re-replication of heterochromatin like in *atxr5/6* (see new supplementary Figure 14d,e).

As requested by reviewer #2, we have analyzed nuclear localization of MAIL1 and MAIN proteins using GFP-fusion proteins. This analysis suggests that MAIL1 and MAIN are not preferentially located at chromocenters (see new supplementary figure 15). However, following reviewer #3's recommendation, we have performed FISH experiments on *mail1*, *main*, *mail1 main* and wild-type nuclei using a probe to the 106 pericentromeric repeats (see new Figure 5). These experiments nicely support the conclusion that *MAIL1* and *MAIN* play a role in condensation of pericentromeric heterochromatin.

4. The logic regarding the phylogenetic claims appear rational, however, this reviewer defers to other reviewers with more expertise in the area to fully evaluate the authors claims.

We have significantly extended our phylogenetic analyses in response to points raised by reviewer #4.

Reviewer #4 (Remarks to the Author):

The manuscript by Mathieu et al. evidences a new pathway for gene(TE)-silencing in plants, which seems to represent a great advance in the plant research domain. I trust the works done being quite solid, based on the genetic and NGS analyses, but I admit I'm not competent in plants genetics, and then not competent to foresee the impact of the paper in the future.

As a general comment, I'm surprised by the shortness of the manuscript (about 1500 words for main text and 1700 for methods, and 3 figures only, while, if I'm not wrong, Nature Communications allows up to 5000 words in the main text and 3000 in the methods, and up to 10 figures. It is for sure that the huge amount of work done would deserve a more developed manuscript, and I would advice the authors to profit of this opportunity to better detail their work. For now it sounds like a big summary, which leaves the reader a bit frustrated...

The manuscript was transferred from another journal, which had different formatting requirements, and we have been informed that it was not necessary to reformat the manuscript before consideration at Nature Communications. The revised version of the manuscript has been significantly extended, while addressing the different points raised by the reviewers.

The part about evolution, I have been asked to review, would of course benefits from this manuscript expansion. Yet I have the feeling it is not the main groundbreaking point of the manuscript (although the cooptation notion is in the title, in this way, the title may not be adapted unless some more work is done on the evolution of the domain). In my opinion, efforts should be kept for expanding a bit the genetic part. Anyway I believe the last part on domain evolution could be improved by adding some more methodological details, and some other modifications. If the title has to be kept, a real effort

would be necessary to reinforce this part of the manuscript, which for now, appears like a very small conclusive paragraph. In particular, the hypothesis of an exaptation by the genome of a TE protein (as opposed to the alternative hypothesis of a capture of genomic sequence by transposons) should be better supported by a more in-depth analysis, if possible.

Below are my specific comments on this part:

Point 1:

I don't see any argument to state that the PMD-A clade (gypsy) is ancestral. In absence of any outgroup, a phylogenetic tree cannot be rooted, and the PhyML proposed topology is totally arbitrarily, unless you specify which of the sequences are outgroup (in some phyML versions, specification is done by adding a * to the sequence name.). I'm not sure this has been done, and if so, I would expect some arguments for choosing PMD-A as the outgroup, rather than another one. In any other case, the tree should be kept as unrooted. This point is important since it seems to be the initial statement on which the further evolutionary scenario is based.

We agree that there is no way at the moment to efficiently root this tree (as no clear outgroup is known) and the tree presented in Figure 6 should be considered as unrooted. So, it is correct to say that we do not have at the moment decisive arguments to propose PMD-A as ancestral to PMD-B and C. However, our phylogenetic analysis gives us some arguments to propose that the association of PMD-A with Gypsy elements happened much earlier in evolution compared to the association of PMD-C with MULE. First, PMD-A forms two well-supported clades distinct from PMD-B and C. This is not the case of MULE associated PMDs that are deeply embedded within the PMB-C clade. Next Gypsy containing PMDs are distributed in a large number of dicot and monocot species (16 of the 33 species used, including the ancestral *Amborella* species) while the association of PMD-C with MULE is limited to a few grass species and *Amborella*. Furthermore, these MULE-associated PMD-C are closely related to their genic versions (a situation very different for PMD-A that are much more divergent from the genic PMDs) suggesting a recent capture event in this case. It is intriguing to find also in *Amborella* an association of PMD-C with MULE, but here again this TE-associated version is very close to the *Amborella* genic version (and quite divergent from the grass MULE-associated PMD-C). In conclusion, we feel we have strong argument to propose a scenario in which the PMD-MULE association result of two independent captures of a genic PMD by a MULE, one in the grass lineage and one very recently in the *Amborella* lineage and that the PMD-Gypsy association is more ancestral to these events.

To explain the large distribution of the PMD-A Gypsy in monocot and dicot species, we propose that this association took place in the common ancestor of angiosperms. The fact that PMD-A Gypsy elements are not found in all angiosperm species is not in contradiction with this proposition. Indeed, the evolutionary success of a given transposable element family can vary from one lineage to another. In our case, PMD-A containing Gypsy elements managed to amplify successfully (and recently enough to be observable) in the *Amborella* lineage, in the grass lineage as well as in the Fabidae, the Malpighiales, the Solanales and the Vitales lineages. However, with the exception of *Theobroma cacao*, it became extinct in the *Brassicales* lineages.

Now the question whether PMD-A is ancestral or not to PMD-B and C still remains. Two possibilities exist. A Gypsy element may have co-opted a genic PMD in the angiosperm common ancestor or a genic PMD may have been domesticated in the angiosperm common ancestor from a Gypsy element. We take into account these two possibilities in the revised manuscript.

Point 2:

I regret that, while obvious efforts have been done to choose representative (and "basal") species of dicot and monocot, the tree in the figure does not highlight the species distribution. I would suggest adding some indications about at least the origin (angiosperm, monocot or dicot) of each sequence (by marks or colors...). Discussion about this could also be interesting, since it can give indication

about the time at which suspected events (exaptation, recapture...) took place. (Here I suspect it may be difficult, but it is worth to try.)

The new tree presented in Figure 6 has been color coded for simplicity (green for dicot species, orange for monocot species, blue for the basal *Amborella* species and yellow for the two species basal to the grass lineage). For the timing of the capture/domestication events, see our answer to point 1.

Point 3:

It seems clear that few efforts have been done previously to characterize the PMD domain (Babu et al. (2006) actually focused on another domain, and the PMD description is just a by-product of their analysis). However, there is another reference dealing a bit with this PMD domain in plants gypsy-like elements (Steinbauerova et al 2011 Genetica) that could be cited.

We now cite this paper in the revised manuscript.

Point4:

Furthermore, although not that easy to find, the PMD domain is already defined in the pfam database (PF10536), based on similarities found on several thousands of sequences (<http://pfam.xfam.org/family/PF10536#tabview=tab3>). A full set of 2700 aligned sequences, or a representative set of 65 sequences are available for downloading. These sequences come from the uniprot database, and have been identified through a HMM-based strategy. I think it is a great idea to identify this domain directly from the genome DNA sequences, even if the BLAST (TBLASTN) methodology is probably less efficient than HMM. However, by restricting the search to the 52 genomes of the phytozome, or to the 9 selected species, some interesting data may be lost. This huge amount of data in the pfam database should be exploited in order to gain support to the proposed scenario. I would suggest the authors to explore this huge dataset, to refine their statement as well as to confirm their phylogenetic analysis.

From what I tried (raw phylogenetic analysis of the 2700 sequences, and phylogenetic analysis of the 65 representatives sequences + the authors sequences), the phylogeny of the authors is of course a simplification but globally stay in agreement with the tree obtained with the full set of sequences. Again, this dataset would provide great materials for a deeper/wider analysis of the domain.

As suggested, we have significantly expanded our data set to build the tree presented in the new Figure 6, from 9 species (and 65 proteins) to 33 species (and 308 proteins). To make sure that our selected sequences are truly representative of the PMD diversity present in each species, we first collected all PMD proteins not associated to transposable elements as well as TE-associated PMDs representing the different TE families within each species. This led us to collect initially 464 proteins from the 33 species. We next performed a preliminary phylogenetic analysis of PMDs independently for each species. From these individual analyses, we were able to select PMDs that are truly representative of the diversity within each of our 33 species leading to the 308 proteins that were aligned and used to build our final phylogenetic tree. We did not use directly the pfam database for several reasons. First, although the pfam database covers 47 species, it is less representative of the angiosperm diversity compared to our selected 33 species (for example the pfam database contains 9 species of the *Oryza* genus but only two species of the very large Fabidae clade (from which we have used 7 species). It also contains a very large number of PMDs for some species (for example 491 PMDs only from *Oryza* species, 273 from *Jatropha cursea*) but only 5 PMD containing proteins from *Cucumis sativus* (although we could find 8 PMDs in this species, 5 of them were included as representative in the final analysis). This database is also composed of several hundred PMD motifs associated with MULE elements but curiously contains almost no PMD motif associated to Gypsy elements. For all these reasons, we have decided to build our own representative database.

Point 5:

The authors propose a scenario in which the PMD domain was first present in Gypsy elements, then has been exapted by genomes and has diverged into two groups, before some sequences from one

group being captured by MULEs elements. Noticeably, this scenario is the opposite of the one proposed in Steinbauerova et al 2011 *Genetica*, who suggest the PMD domain was transmitted to gypsy elements after insertion of a MULE element within a gypsy element.

I have to say that, based on the arguments presented here and there, I'm not fully convinced by any of these scenarios (although one of them should of course approach the truth...). I found anyway an interesting remark in Steinbauerova et al 2011: The identified PMD-carrying Gypsy elements (Tat group) are relatively recent elements (identified by LTR-Harvest if I remember well), meaning ancient elements are not identified and then cannot be used to date the insertion event. In the present manuscript, the phylogenetic analysis, compared to species phylogeny, could bring some useful information. The fact that only recent Gypsy elements are identified should also be taken into account for the discussion.

Please see our answer to point 1 for a justification of our supported scenarios

Point 6:

It is said that the PMD-A is exclusively found in Gypsy elements. There is no detail in the method part of how the presence of gypsy sequences on both sides of the PMD-A domains have been verified. From what I saw, rice sequences are annotated as Gypsy, but I could not find annotation for the other species. Of course it is probably because I'm not familiar with phytozome website. However I think it would be valuable to mention whether the information is drawn from annotation files for the species, or from independent checking of the flanking sequences. My underlying question is whether this clade is really exclusively found in Gypsy elements (Tat group?). I suppose so for Os and Md sequences, based on annotation or on the fact they are highly repeated in the genome. Yet, I'm not even sure for Zm9 and Sb sequences (Curiously, for Zm9, I was even unable to retrieve the real sequence on the Zea mays genome at the phytozome or at NCBI!). Here also, checking on the data provided by the pfam database may help to support the assertion that PMD-A is exclusively made of gypsy sequences.

The same remarks stand for the two other clades, and the presence or absence of TE-carried PMD elements, in particular, the absence of any TE among the PMD –B clade. Again, pfam data would be useful, along with own verifications about the fact that sequences are part of TEs or not, within each clade.

To confirm the association of a PMD motif with a TE we first used the Repbase database (<http://www.girinst.org/>) to identify annotated Gypsy or Mule elements corresponding to our sequences. PMD-containing Gypsy and PMD-containing Mule described in Repbase and present in our selected species were collected and included in our phylogenetic tree (Repbase names in parenthesis in Figure 6). These sequences were used as references to identify putative new TE-associated PMDs in our phylogenetic analysis. Each time a PMD cluster in our phylogenetic analysis with at least one annotated TE, the genomic sequences corresponding to this PMD was collected and translated (using Fgenesh available at <http://www.softberry.com>) to confirm the association. To test the non-association of "genic" PMD-B and PMD-C, the genomic region of these motifs was submitted to CENSOR, (<http://www.girinst.org/censor/index.php>) the repeat masking tool of Repbase, to make sure that this coding sequence was not embedded within a transposable element. The Material and Methods section of the paper was modified to include this procedure.

For information, in the initial version of the paper, the Zm9 sequence was discovered following a tblastn analysis of a maize EST database (and cannot be retrieved by a general blast analysis as it does not include EST).

Point 7:

I will admit that the details I'm asking for will never be sufficient to be sure about the proposed scenario. Besides the data provided, beside the improvement that could be brought, there is also the big problem of horizontal transfers, frequent for TEs, that can blur the results, making any scenarios

always speculative, unless hardcore TE phylogeny analysis is done in parallel... (yet I don't think it is the point here).

This does not mean that the scenario is not plausible, but just that the authors should be more cautious when stating that "this study demonstrates that MAIL1 and MAIN,...., and that evolved from a TE –encoded pathway via domestication of a TE protein.".

Our phylogeny does not point to horizontal transfer, as most PMDs present in TEs do group in the tree according to their species of origin. Also, the wide distribution of PMD-containing Gypsy elements in numerous monocot and dicot species would necessitate a very large number of horizontal transfer events, which seems unlikely compare to the presence of these Gypsy elements in the angiosperm common ancestor. In accordance with what we wrote in our answer to point 1, we have modified our text to indicate the difficulties to decide which scenario could be the correct one.

Point 8:

Finally, also concerning the conclusion, although this time, the authors precise that it is a speculation, I would like to pinpoint the fact that the presence of some protein domains within TEs is not at all an evidence of a functional role, nor the fact that this domain is frequently found in TEs (since non functional sequences can just be amplified as a consequence of TE amplification). Here it is obvious that the functional role (within TE) is far from elucidated, although the work on MAIN and MAIL1 give some important insights on the role this domain may have.

We fully acknowledge that further studies are needed to assess whether TE-associated PMDs play a functional role in TE life cycles. We only mention that this is an attractive possibility in the discussion section.

I acknowledge that my recommendations imply some substantial work, and there is no warranty that any better-supported evolutionary scenario may emerge. Yet I believe it would greatly improve this part of the manuscript, and hopefully gives more support to the proposed title of the manuscript.

Reviewers' comments:

Reviewer #2 (Remarks to the Author):

The authors claim that MAIN and MAIL1 have a novel function. The mutants appear to have defective organization of chromosomes within the nucleus, which perhaps affects the expression level of genes and TEs (notably, although the manuscript is written focused on TEs, Fig 2 indicates many more protein-coding genes than TE change in expression in mail1 mutants). In the absence of additional understanding of the molecular function of MAIL and MAIN, whether this can be called a novel mechanism remains debatable.

Overall, mail1 mutants have limited effects, influencing the expression of only a handful of genes and TEs. Yet, many of the analyses are geared towards examining large genome-wide effects. The authors have responded to our previous comment about DMRs. However, the statement that there is no methylation difference between mail1 and WT seems a little inaccurate as DMRs are observed. The low overlap between DMRs and mis-regulated genes might be the result of them looking for DMRs that at least overlap the gene by 20%. The comparison with ddm1 is not suitable because ddm1 mutants, unlike mail1 mutants, have large effects on the genome.

Finally, the conclusion that the PMD domain originated in Gypsy TEs, then became a protein-coding gene, and then became part of a MULE TE does not seem any better supported in this version of the manuscript than in the previous version. There is no convincing evidence to support the statement in the last sentence of the abstract "Our results reveal that a genic version of a TE-related domain was used early in plant evolution to enforce silencing, and that this domain was latter (sic) on recaptured by a different class of TE." Even if the authors could demonstrate this, the biological or evolutionary relevance of such a series of events is not clear. There is also no data to indicate the function of the domain early in plant evolution, or to indicate that this domain is part of a tug-of-war between TEs and hosts.

Minor:

Sup Fig 3C is mislabeled. mail1-1 and mail1-2 are switched.

Reviewer #3 (Remarks to the Author):

In their revised manuscript, Ikeda et al. satisfactorily addressed most of my comments through the inclusion of additional data and discussion. This new data further strengthens the author's claims regarding the roles of the MAIL1 and MAIN proteins in gene silencing. Besides some minor comments (listed below) my only remaining concern is the interpretation of the relationship between MAIL, MAIN and the ATXRs. However, I feel my concerns regarding this point would be adequately address through additional of a few experiments in the mail1atxr5atxr6 triple mutant as discussed below.

Major comments:

The authors claim that MAIL1 and MAIN function through a novel pathway, yet they affect similar repressive histone marks as AXTR5/AXTR6—namely, H3K27me1 marks. While it is true that the mail1atxr5atxr6 triple releases silencing at more loci than the mail1 or atxr5atxr6 mutants, the atxr5atxr6 mutant is not a null. Thus, it is still possible that the triple represents the enhancement of weaker mutants. For the other pathways (DNA methylation and siRNAs) the mutants used for the genetic tests have very strong phenotypes and specific example loci were examined to show that at sites that completely lose methylation, for example in the ddm1 mutant, the silencing phenotype is increased when a mail1 mutation is added. Based on these findings the authors have very strong support that MAIL1 acts independently of DNA methylation and siRNAs, but the data regarding links to H3K27me1 pathways is weaker.

To look further into the connection with ATXR5 and ATXR6, the authors also looked at effects on endoreduplication and see no effect in the mail1 single. However, it is not clear if the atxr5 or atxr6 single mutants, which have a weak effect on H3K27me1 levels, show endoreduplication defects. Without looking at H3K27me1 levels (by ChIP qPCR at mail1 affected loci; see Fig S11c) and endoreduplication defects in the mail1atxr5atxr6 to see if the phenotypes are enhanced relative to the mail1 and atxr5atxr6 mutants it remains unclear whether these are completely independent pathways. If the atxr5atxr6 phenotypes are enhanced then these factors will be new proteins important for ATXR mediated silencing—a pathway that we understand very little about—and if they fail to show enhancement then this will suggest MAIL1 and MAIN act in an independent pathway. Either outcome would be significant and of broad interest to the field.

Minor comments:

Methods:

-) It is not clear from the figures or the methods how euchromatic vs heterochromatic regions are determined. This information should be included to allow full interpretation of the data.

-) For the reanalyzed RNA seq data in the mom1 and atxr56 mutants additional details should be added to the methods include the following information:

1. What tissues were used for these RNAseq experiments?
2. How many replicates were included?
3. Was the data processing and DEG analysis the same as described in the methods for the other RNAseq analyses?

-) For the RNA seq data, the methods indicated 2-3 week old seedlings were used. Please clarify what is meant by this: multiple collections over a 1 week period, different stages for rep1 vs rep2, different stages for different mutants?

Fig S4b: Several of the colors (for example the two yellows) are hard to distinguish. Perhaps adding a pattern, like diagonal stripes, would be helpful.

Fig 3a: The dashed lines are difficult to see. Please consider increasing the line thickness.

Fig S9a: The signal for the siRNAs in the northern blots shown is quite weak, making it

difficult to determine if the levels are similar in the wt and mail1 mutants. Since smRNAseq experiments were also conducted in these backgrounds, addition of screenshots showing the normalized 24nt siRNA levels at these loci in the wt and mail1 mutants should also be included.

Fig 5a: Please include representative examples of the three classifications of FISH staining for Wt, mail1, main and the mail1main double mutants.

Reviewer #4 (Remarks to the Author):

The revised version of manuscript 107258-1 by Ikeda et al. has greatly improved and is much clearer for me.

In particular, the authors have satisfactorily answered to all the points raised previously. Concerning the last part on the PMD evolution, and relationship with TEs, the authors have removed all statements that were not supported enough and objectively propose now alternative scenarios. The phylogenetic analysis includes more species and although the tree is more complicated, the color code helps to understand how this domain has evolved in angiosperms.

Some remarks:

- It is known that MULE elements are able to capture genomic sequence including gene fragments (a process called transduplication) although the capture fragment is often non functional (works of T. E. Bureau, and some others...)
- The method for identification of TE-associated PMD is still confusing (line 457-459, p. 21 un the methods section, and next sentence too). I understand that the nucleotide (genomic) sequences corresponding to PMDs clustered with a MULE-PMD are translated to confirm the association. I suppose that the authors want to refer to the surrounding genomic sequence and not the PMD sequence itself?
- The two last sentences in the abstract appear redundant.

Reviewers' comments:

Reviewer #2 (Remarks to the Author):

The authors claim that MAIN and MAIL1 have a novel function. The mutants appear to have defective organization of chromosomes within the nucleus, which perhaps affects the expression level of genes and TEs (notably, although the manuscript is written focused on TEs, Fig 2 indicates many more protein-coding genes than TE change in expression in mail1 mutants). In the absence of additional understanding of the molecular function of MAIL and MAIN, whether this can be called a novel mechanism remains debatable.

As stated by reviewer #3, we believe our data provide “very strong support that MAIN and MAIL1 act independently of DNA methylation and siRNAs”. We further show that H3K9me2 is not altered in *mail1* and *main* mutants. Additionally, we show that, although mutations in *MAIL1* and *MAIN* lead to decreased H3K27me1 levels, this is much likely not the cause of silencing release. Our genetic analyses also provide strong evidence of MAIL1 acting in a molecular pathway largely independent of MOM1 and AtMORCs, which are involved in the two best described silencing pathways independent of DNA methylation in Arabidopsis. Together, our data strongly suggest that *MAIL1/MAIN* are involved in a previously unidentified silencing pathway. Nonetheless, we do not use the adjective “novel” when referring to *MAIL1/MAIN* silencing pathway in the new revised manuscript.

Overall, *mail1* mutants have limited effects, influencing the expression of only a handful of genes and TEs. Yet, many of the analyses are geared towards examining large genome-wide effects.

Our transcriptome analyses were performed applying very stringent cutoffs to define differentially expressed loci (fold change ≥ 4 and FDR < 0.01 in both replicates), and identified a total of 39 TEs (36 up-regulated) and 358 genes mis-regulated in *mail1*. We do not share reviewer’s opinion that these are limited effects. For comparison purposes, using similarly stringent thresholds, 52 and 19-26 TEs were defined as upregulated in *mom1-2* and *atmorc6* silencing mutants, respectively (Moissiard et al. 2012. Science; Moissiard et al. 2014. PNAS). Our manuscript includes a balance of both genome-wide (RNA-seq, BS-seq, siRNA-seq) and locus-specific analyses (RT-qPCR, siRNA Northern blots, ChIP) together with FISH and cytological analyses, which, we think, allows for a precise description of the mutants’ molecular phenotypes.

The authors have responded to our previous comment about DMRs. However, the statement that there is no methylation difference between *mail1* and WT seems a little inaccurate as DMRs are observed. The low overlap between DMRs and mis-regulated genes might be the result of them looking for DMRs that at least overlap the gene by 20%. The comparison with *ddm1* is not suitable because *ddm1* mutants, unlike *mail1* mutants, have large effects on the genome.

We did not claim that “there is no methylation difference between *mail1* and WT”. It is stated: “DNA methylation with functional consequences is largely unaltered in *mail1*” (page 6, line 7). Indeed, our analysis allowed detection of a few DMRs in *mail1*, in a similar range as in *mom1* and *atmorc6* mutants, and they poorly overlapped with *mail1* up-regulated loci. Following reviewer’s comment, we have re-analyzed the overlap between hypomethylated DMRs and upregulated TEs/PCGs by decreasing the minimum required overlap to only one single nucleotide of the DMR. Changing this parameter did not enhance the overlap between DMRs and upregulated loci, confirming that transcriptional upregulation in *mail1* is largely independent of changes in DNA methylation. The methods section has been modified accordingly. The *ddm1* mutant data were included in our analysis as a positive control for our DMR calling pipeline; we make no comparison with this mutant in the revised manuscript.

Finally, the conclusion that the PMD domain originated in Gypsy TEs, then became a protein-coding gene, and then became part of a MULE TE does not seem any better supported in this version of the manuscript than in the previous version.

We respectfully disagree with this comment. There is no way to formally demonstrate an evolution scenario other than supporting this scenario with clear and relevant phylogenetic data such as those we provided in our revised manuscript. To our opinion, our phylogenetic analysis provides strong support to the conclusions and proposed scenarios – an opinion that is shared by reviewer 4, whom states: “Concerning the last part on the PMD evolution, and relationship with TEs, the authors have removed all statements that were not supported enough and objectively propose now alternative scenarios”.

There is no convincing evidence to support the statement in the last sentence of the abstract “Our results reveal that a genic version of a TE-related domain was used early in plant evolution to enforce silencing, and that this domain was latter (sic) on recaptured by a different class of TE.” Even if the authors could demonstrate this, the biological or evolutionary relevance of such a series of events is not clear. There is also no data to indicate the function of the domain early in plant evolution, or to indicate that this domain is part of a tug-of-war between TEs and hosts.

The last sentence of the abstract and the last sentence of the discussion have been removed.

Minor:

Sup Fig 3C is mislabeled. *mail1-1* and *mail1-2* are switched.

Thank you for pointing this mistake out. This is now corrected.

Reviewer #3 (Remarks to the Author):

In their revised manuscript, Ikeda et al. satisfactorily addressed most of my comments through the inclusion of additional data and discussion. This new data further strengthens the author's claims regarding the roles of the MAIL1 and MAIN proteins in gene silencing. Besides some minor comments (listed below) my only remaining concern is the interpretation of the relationship between MAIL, MAIN and the ATXR5s. However, I feel my concerns regarding this point would be adequately addressed through additional of a few experiments in the mail1atxr5atxr6 triple mutant as discussed below.

Major comments:

The authors claim that MAIL1 and MAIN function through a novel pathway, yet they affect similar repressive histone marks as AXTR5/AXTR6—namely, H3K27me1 marks. While it is true that the mail1atxr5atxr6 triple releases silencing at more loci than the mail1 or atxr5atxr6 mutants, the atxr5atxr6 mutant is not a null. Thus, it is still possible that the triple represents the enhancement of weaker mutants. For the other pathways (DNA methylation and siRNAs) the mutants used for the genetic tests have very strong phenotypes and specific example loci were examined to show that at sites that completely lose methylation, for example in the ddm1 mutant, the silencing phenotype is increased when a mail1 mutation is added. Based on these findings the authors have very strong support that MAIL1 acts independently of DNA methylation and siRNAs, but the data regarding links to H3K27me1 pathways is weaker.

To look further into the connection with ATXR5 and ATXR6, the authors also looked at effects on endoreduplication and see no effect in the mail1 single. However, it is not clear if the atxr5 or atxr6 single mutants, which have a weak effect on H3K27me1 levels, show endoreduplication defects. Without looking at H3K27me1 levels (by ChIP qPCR at mail1 affected loci; see Fig S11c) and endoreduplication defects in the mail1atxr5atxr6 to see if the phenotypes are enhanced relative to the mail1 and atxr5atxr6 mutants it remains unclear whether these are completely independent pathways. If the atxr5atxr6 phenotypes are enhanced then these factors will be new proteins important for ATXR mediated silencing—a pathway that we understand very little about—and if they fail to show enhancement then this will suggest MAIL1 and MAIN act in an independent pathway. Either outcome would be significant and of broad interest to the field.

Work from the Jacobsen and Michaels labs previously showed that atxr5 and atxr6 single mutants do not display endoreduplication defects (Jacob et al. 2010. Nature; Ref. 4 in our manuscript).

Following reviewer's request, we have analyzed endoreduplication and H3K27me1 levels in mail1 atxr5 atxr6 triple mutants. Data from these two analyses reveal no enhancement of corresponding atxr5 atxr6 molecular phenotypes in mail1 atxr5 atxr6 triple mutants relative to atxr5/6 and/or mail1 (Supplementary Fig. 13b and Supplementary Fig. 14e in the new revised manuscript). These additional experimental data provide further support for our conclusion that MAIL1 and ATXR5/6 act in largely separated molecular pathways.

Minor comments:

Methods:

-) It is not clear from the figures or the methods how euchromatic vs heterochromatic regions are determined. This information should be included to allow full interpretation of the data.

Genome-wide coordinates of pericentromeric heterochromatin vs euchromatin were assigned based on the distribution of repetitive elements, protein-coding genes and DNA methylation across chromosomes as described previously (Bernatavichute et al. 2008; Ref. 2 in our manuscript). This is now mentioned in the legend of Figure 2b of the new revised manuscript.

-) For the reanalyzed RNA seq data in the mom1 and atxr56 mutants additional details should be added to the methods include the following information:

1. What tissues were used for these RNAseq experiments?
2. How many replicates were included?
3. Was the data processing and DEG analysis the same as described in the methods for the other RNAseq analyses?

We have included these details in the "RNA sequencing" paragraph of the methods section in the new revised manuscript.

-) For the RNA seq data, the methods indicated 2-3 week old seedlings were used. Please clarify what is meant by this: multiple collections over a 1 week period, different stages for rep1 vs rep2, different stages for different mutants?

We used 18-day-old seedlings (2.5 weeks) for all RNA-seq samples (and BS-seq samples). This is clarified in the new revised manuscript.

Fig S4b: Several of the colors (for example the two yellows) are hard to distinguish. Perhaps adding a pattern, like diagonal stripes, would be helpful.

We have tried using patterns for this figure, but we feel this does not really help. We have modified colors to try helping the reader. Also, please note that the TE superfamilies are listed in the same order in the histogram plot and in the color key.

Fig 3a: The dashed lines are difficult to see. Please consider increasing the line thickness.
We have colored these lines in black.

Fig S9a: The signal for the siRNAs in the northern blots shown is quite weak, making it difficult to determine if the levels are similar in the wt and mail1 mutants. Since smRNAseq experiments were also conducted in these backgrounds, addition of screenshots showing the normalized 24nt siRNA levels at these loci in the wt and mail1 mutants should also be included.
As requested, we have included genome browser screenshots of normalized 21- and 24-nt siRNA levels along the loci analyzed by RNA gel blots (Supplementary Fig 9c of the new revised manuscript).

Fig 5a: Please include representative examples of the three classifications of FISH staining for Wt, mail1, main and the mail1main double mutants.
Representative images have been included in the new revised manuscript (new Supplementary Fig. 15)

Reviewer #4 (Remarks to the Author):

The revised version of manuscript 107258-1 by Ikeda et al. has greatly improved and is much clearer for me. In particular, the authors have satisfactorily answered to all the points raised previously. Concerning the last part on the PMD evolution, and relationship with TEs, the authors have removed all statements that were not supported enough and objectively propose now alternative scenarios. The phylogenetic analysis includes more species and although the tree is more complicated, the color code helps to understand how this domain has evolved in angiosperms.
Thank you.

Some remarks:

- It is known that MULE elements are able to capture genomic sequence including gene fragments (a process called transduplication) although the capture fragment is often non functional (works of T. E. Bureau, and some others...)

Thank you for pointing this out. This is now mentioned in the discussion section of the revised manuscript.

- The method for identification of TE-associated PMD is still confusing (line 457-459, p. 21 un the methods section, and next sentence too). I understand that the nucleotide (genomic) sequences corresponding to PMDs clustered with a MULE-PMD are translated to confirm the association. I suppose that the authors want to refer to the surrounding genomic sequence and not the PMD sequence itself?

Yes, we intended to refer to the surrounding genomic sequence and not to the PMD. We have modified this part of the Methods for clarification.

- The two last sentences in the abstract appear redundant.

The last sentence of the abstract has been removed.

REVIEWERS' COMMENTS:

Reviewer #2 (Remarks to the Author):

Removal of the last sentence of the abstract improves the paper.

The authors have retained the argument that the PMD domain is domesticated from a TE. What is the evidence for this?

- 1) The plant mobile domain in clades A1 and A2 are associated with Gypsy retrotransposons.
- 2) In clade C, monocot PMDs and 1 Amborella PMD are associated with MULE elements.

What remains unknown/lacking:

- 1) The relationship between clade A1 and clade C, of which MAIN and MAIL are members. In the absence of such knowledge, the capture of PMD-C by MULE in a few species hardly seems remarkable; MULEs have captured many genic fragments.
- 2) Any evidence that the PMD functions in the TE life cycle in the instances where it is fused to a TE, for either gypsy retrotransposons or MULEs.
- 3) Any evidence that the PMD domain of MAIN or MAIL are specifically responsible for the functions proposed here.
- 3) Notably, MAIN and MAIL are not part of TE fusions.

Reviewer #3 (Remarks to the Author):

In their revised manuscript, Ikeda et al. satisfactorily addressed all of my previous comments through the inclusion of additional data and discussion. This new data further strengthens the author's claims regarding the roles of the MAIL1 and MAIN proteins and their participation in a novel gene silencing pathway that is independent of all known silencing pathways. The identification of a new pathway acting largely downstream of DNA methylation provides additional insights into how DNA methylation may affect gene expression—a key aspect of DNA methylation that has remained unclear. As such, this work marks a major advance and will be of broad interest within the areas of gene regulation and epigenetics.

REVIEWERS' COMMENTS:

Reviewer #2 (Remarks to the Author):

Removal of the last sentence of the abstract improves the paper.

The authors have retained the argument that the PMD domain is domesticated from a TE. What is the evidence for this?

- 1) The plant mobile domain in clades A1 and A2 are associated with Gypsy retrotransposons.
- 2) In clade C, monocot PMDs and 1 Amborella PMD are associated with MULE elements.

What remains unknown/lacking:

- 1) The relationship between clade A1 and clade C, of which MAIN and MAIL are members. In the absence of such knowledge, the capture of PMD-C by MULE in a few species hardly seems remarkable; MULEs have captured many genic fragments.

We have removed the word “surprisingly” when referring to the PMD capture by *MULE* elements. The ability of *MULE* elements to capture genomic fragments was mentioned in the manuscript: “*MULE* elements are able to capture genomic sequences, including gene fragments, in a process termed transduplication”.

- 2) Any evidence that the PMD functions in the TE life cycle in the instances where it is fused to a TE, for either gypsy retrotransposons or MULEs.

This is correct, and will be the matter of future studies.

- 3) Any evidence that the PMD domain of MAIN or MAIL are specifically responsible for the functions proposed here.

The PMD domain is the only protein domain encoded by *MAIL1* and *MAIN*, and it comprises most of the length of the *MAIL1* and *MAIN* proteins. Nonetheless, we now indicate that “the function of the PMD domain itself remains to be determined” in the last sentence of the discussion of the revised manuscript.

- 3) Notably, MAIN and MAIL are not part of TE fusions.

This is correct.

Reviewer #3 (Remarks to the Author):

In their revised manuscript, Ikeda et al. satisfactorily addressed all of my previous comments through the inclusion of additional data and discussion. This new data further strengthens the author's claims regarding the roles of the *MAIL1* and *MAIN* proteins and their participation in a novel gene silencing pathway that is independent of all known silencing pathways. The identification of a new pathway acting largely downstream of DNA methylation provides additional insights into how DNA methylation may affect gene expression—a key aspect of DNA methylation that has remained unclear. As such, this work marks a major advance and will be of broad interest within the areas of gene regulation and epigenetics.

Thank you.